# CLIQUEFORMER: MODEL-BASED OPTIMIZATION WITH STRUCTURED TRANSFORMERS

## ABSTRACT

Expressive large-scale neural networks enable training powerful models for prediction tasks. However, in many engineering and science domains, such models are intended to be used not just for prediction, but for design—e.g., creating new proteins that serve as effective therapeutics, or creating new materials or chemicals that maximize a downstream performance measure. Thus, researchers have recently grown an interest in building deep learning methods that solve offline *model-based optimization* (MBO) problems, in which design candidates are optimized with respect to surrogate models learned from offline data. However, straightforward application of predictive models that are effective at predicting in-distribution properties of a design are not necessarily the best suited for use in creating new designs. Thus, the most successful algorithms that tackle MBO draw inspiration from reinforcement learning and generative modeling to meet the in-distribution constraints. Meanwhile, recent theoretical works have observed that exploiting the structure of the target black-box function is an effective strategy for solving MBO from offline data. Unfortunately, discovering such structure remains an open problem. In this paper, following first principles, we develop a model that learns the structure of an MBO task and empirically leads to improved designs. To this end, we introduce *Cliqueformer*—a scalable transformer-based architecture that learns the black-box function's structure in the form of its *functional graphical model* (FGM), thus bypassing the problem of distribution shift, previously tackled by conservative approaches. We evaluate Cliqueformer on various tasks, ranging from high-dimensional black-box functions from MBO literature to real-world tasks of chemical and genetic design, consistently outperforming the baselines.

## 1 INTRODUCTION

Most of the common use cases of deep learning (DL) so far have taken the form of prediction tasks (Hochreiter & Schmidhuber, 1997; He et al., 2016; Krizhevsky et al., 2017; Vaswani et al., 2017). However, in many applications, *e.g.* protein synthesis or chip design, we might want to use powerful models to instead solve *optimization* problems. Clearly, accurate predictions of a target score of an object could be used to find a design of the object that maximizes that score. Such a methodology is particularly useful in engineering problems in which evaluating solution candidates comes with big risk. For example, synthesizing a proposed protein requires a series of wet lab experiments and induces extra cost and human effort (Gómez-Bombarelli et al., 2018; Brookes et al., 2019). Thus, to enable proposing *de-novo* generation of strong solution candidates, researchers have drawn their attention to offline *black-box optimization* (BBO), often referred to as *model-based optimization* (MBO). In this paradigm, first, a surrogate model of the score is learned from offline data. Next, a collection of designs is trained to maximize the surrogate, and then proposed as candidates for maximizers of the target score (Gómez-Bombarelli et al., 2018; Kumar et al., 2021).

Unfortunately, model-based optimization (MBO) introduces unique challenges not encountered in classical prediction tasks. The most significant issue arises from the incomplete coverage of the design space by the data distribution. This limitation leads to a phenomenon known as *distribution shift*, where optimized designs drift away from the original data distribution. Consequently, this results in poor proposals with significantly overestimated scores (Trabucco et al., 2022; Geng,

2023). To address it, popular MBO algorithms have been employing techniques from offline reinforcement learning (Kumar et al., 2020; Trabucco et al., 2021) and generative modeling (Kumar & Levine, 2020; Mashkaria et al., 2023) to enforce the in-distribution constraint. Meanwhile, much of the recent success of DL has been driven by domain-specific neural networks that, when scaled together with the amount of data, lead to increasingly better performance. While researchers have managed to establish such models in several fields, it is not immediately clear how to do it in MBO. Recent theoretical work, however, has shown that MBO methods can benefit from information about the target function's *structure*, which can be implemented as a decomposition of the surrogate over the target's *functional graphical model* (Grudzien et al., 2024, FGM). This insight opens up new possibilities for developing more effective MBO models by injecting such structure into their architecture. However, how to integrate such decompositions into scalable neural networks remains an open question, and addressing this challenge is the focus of this work.

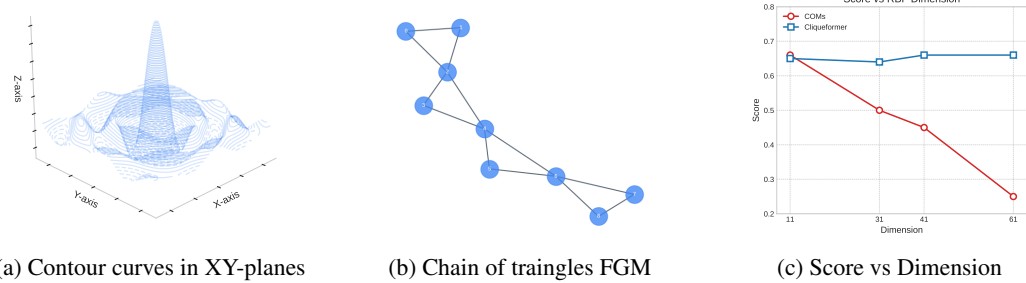

(a) Contour curves in XY-planes    (b) Chain of traingles FGM    (c) Score vs Dimension

Figure 1: The first building block of the LRBF tasks are 3D radial-basis functions (left). These functions are applied to triplets arranged in a chain of triangles FGM (center) and linearly mixed. Then, an observable designs are produced with non-linear transformations of the chain and, together with their values, form a dataset. We show the score (right) of our structure-learning Cliqueformer and structure-oblivious COMs (Trabucco et al., 2021), against the dimension of LRBF functions, modulated only by varying the number of triangles. Cliqueformer, unlike COMs, sustains strong performance across all dimensions. More results in Section 5.

In contrast to previous works, in our paper, we develop a scalable model that tackles MBO by learning the structure of the black-box function through the formalism of functional graphical models. Our architecture aims to solve MBO by *1)* decomposing the predictions over the *cliques* of the function's FGM, and *2)* enforcing the cliques' marginals to have wide coverage with our novel form of the variational bottleneck (Kingma & Welling, 2013; Alemi et al., 2016). However, building upon our Theorem 2, we do not follow Grudzien et al. (2024) during the FGM discovery step, and instead subsume it in the learning algorithm. To enable scaling to high-dimensional problems and large datasets, we employ the transformer backbone (Vaswani et al., 2017). Empirically, we demonstrate that our model, *Cliqueformer*, inherits the scalability guarantees of MBO with FGM (see Figure 1). We further demonstrate its superiority to baselines in a suite of tasks with latent radial-basis functions (Grudzien et al., 2024) and real-world chemical (Hamidieh, 2018) and DNA design tasks (Trabucco et al., 2022; Uehara et al., 2024).

## 2 PRELIMINARIES

In this section, we provide the necessary background on model-based optimization. Additionally, we cover the basics of functional graphical models on top of which we build Cliqueformer.

### 2.1 MODEL-BASED OPTIMIZATION

We consider a model-based optimization problem, where we are given a dataset $\mathcal{D} = \{\mathbf{x}^i, \mathbf{y}^i\}_{i=1}^N$ of examples $\mathbf{x} \in \mathcal{X}$, following distribution $p(\mathbf{x})$, and their values $\mathbf{y} = f(\mathbf{x}) \in \mathbb{R}$ under an unkown (black-box) function $f : \mathcal{X} \to \mathbb{R}$. Our goal is to optimize this function offline—to find its maximizer

$$\mathbf{x}^\star = \arg\max_{\mathbf{x} \in \mathcal{X}} f(\mathbf{x}) \tag{1}$$

by only using information provided in $\mathcal{D}$ (Kumar & Levine, 2020). Sometimes, a more general objective in terms of a *policy* over $\pi(\mathbf{x})$ is also used, $\eta(\mathbf{x}) = \mathbb{E}_{\mathbf{x} \sim \pi}[f(\mathbf{x})]$. In either form, unlike in

Bayesian optimization, we cannot make additional queries to the black-box function (Brochu et al., 2010; Kumar & Levine, 2020). This formulation represents settings in which obtaining such queries is prohibitively costly, such as tests of new chemical molecules or of new hardware architectures (Kim et al., 2016; Kumar et al., 2021).

To solve MBO, it is typical to learn a model $f_\theta(\mathbf{x})$ of $f(\mathbf{x})$ parameterized by a vector $\theta$ with a regression method and data from $\mathcal{D}$,

$$L(\theta) = \mathbb{E}_{(\mathbf{x},\mathrm{y})\sim\mathcal{D}}\big[\big(f_\theta(\mathbf{x}) - \mathrm{y}\big)^2\big] + \mathrm{Reg}(\theta) \tag{2}$$

where $\mathrm{Reg}(\theta)$ is an optional regularizer. Classical methods choose $\mathrm{Reg}(\theta)$ to be identically zero, while conservative methos use the regularizer to bring the values of examples out of $\mathcal{D}$ down. For example, the regularizer of Conservative Objective Model's (Trabucco et al., 2021, COMs) is

$$\mathrm{Reg}_{\mathrm{com}}(\theta) = \alpha\big(\mathbb{E}_{\mathbf{x}\sim\mu_{\theta_\perp}}[f_\theta(\mathbf{x})] - \mathbb{E}_{\mathbf{x}\sim\mathcal{D}}[f_\theta(\mathbf{x})]\big), \quad \alpha > 0,$$

where $(\cdot)_\perp$ is the stop-gradient operator and $\mu_{\theta_\perp}(\mathbf{x})$ is the distribution obtained with a few gradient ascent steps on $\mathbf{x}$ initialized from $\mathcal{D}$. This distribution depends on the value of $\theta$ but is not differentiated through while computing the loss's gradient, and thus we denote it by $\theta_\perp$. Unfortunately, in addition to the extra computational cost that the inner-loop gradient ascent induces, COMs's regularizer limits the amount of improvement that it allows its designs to make. This is particularly frustrating since recent work on functional graphical models (Grudzien et al., 2024, FGM) delivered a premise of large improvements in the case when the black-box function's graph can be discovered, as we explain in the next subsection.

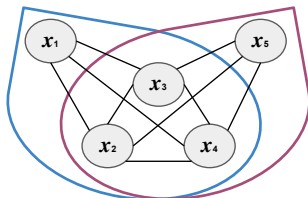

$$f(x) = f_{-5}(x_{-5}) + f_{-1}(x_{-1})$$

Figure 2: An FGM of a 5D function which decomposes as $f(\mathbf{x}) = f_{-5}(\mathbf{x}_{-5}) + f_{-1}(\mathbf{x}_{-1})$. By Definition 1, nodes $\mathrm{x}_1$ and $\mathrm{x}_5$ are not linked.

## 2.2 FUNCTIONAL GRAPHICAL MODELS

An FGM of a high-dimensional function $f(\mathbf{x})$ is a graph over individual components of $\mathbf{x}$ that separates $\mathrm{x}_i, \mathrm{x}_j \in \mathbf{x}$ if their contributions to $f(\mathbf{x})$ are independent of each other. Knowing such a structure of $f$ one can eliminate interactions between independent variables from a model that approximates it, and thus prevent an MBO algorithm from exploiting them. We summarize basic properties of FGMs below. In what follows, we denote $\mathcal{X}_{-i}$ as the design (input) space without the $i^{th}$ subspace, and $\mathbf{x}_{-i}$ as its element [1]. We also write $[K]$ to denote the set $\{1, \ldots, K\}$.

**Definition 1.** *Let $\mathbf{x} = (\mathrm{x}_v \mid v \in \mathcal{V})$ be a joint variable with index set $\mathcal{V}$, and $f(\mathbf{x})$ be a real-valued function. An FGM $\mathcal{G} = (\mathcal{V}, \mathcal{E})$ of $f(\mathbf{x})$ is a graph where the edge set $\mathcal{E} \subset \mathcal{X}^2$ is such that,*

$$\exists\, f_{-i} : \mathcal{X}_{-i} \to \mathbb{R} \text{ and } f_{-j} : \mathcal{X}_{-j} \to \mathbb{R}, \text{ with } f(\mathbf{x}) = f_{-i}(\mathbf{x}_{-i}) + f_{-j}(\mathbf{x}_{-j}), \text{ implies } (i, j) \notin \mathcal{E}.$$

*See Figure 2 for illustration.*

The basic result about FGMs is that they allow for decomposition of the target function into subfunctions with smaller, partially-overlapping inputs, from the FGM's set of maximal cliques $\mathcal{C}$,

$$f(\mathbf{x}) = \sum_{C\in\mathcal{C}} f_C(\mathbf{x}_C). \tag{3}$$

Intuitively, the decomposition enables more efficient learning of the target function since it can be constructed by adding together functions defined on smaller inputs, which are easier to learn. This, in turn, allows for more efficient MBO since the joint solution $\mathbf{x}^\star$ can be recovered by *stitching* individual solutions $\mathbf{x}_C^\star$ to smaller problems. This intuition is formalized by the following theorem.

**Theorem 1** (Grudzien et al. (2024)). *Let $f(\mathbf{x})$ be a real-valued function, $\mathcal{C}$ be the set of maximal cliques of its FGM, and $\Pi$ be a policy class. Let $C_{stat}$ and $C_{cpx}$ be constants that depend on the probability distribution of $\mathbf{x}$ and function approximator class's complexity, respectively, defined in Appendix A. Then, the regret of MBO with the FGM information is given by,*

---

[1]For example, if $\mathcal{X} = \mathcal{X}_1 \times \mathcal{X}_2 \times \mathcal{X}_3$ and $\mathbf{x} = (\mathrm{x}_1, \mathrm{x}_2, \mathrm{x}_3)$, then $\mathcal{X}_{-2} = \mathcal{X}_1 \times \mathcal{X}_3$, and $\mathbf{x}_{-2} = (\mathrm{x}_1, \mathrm{x}_3)$.

$$\eta(\pi^\star) - \eta(\hat{\pi}_{FGM}) \leq C_{stat} C_{cpx} \max_{\pi \in \Pi, \mathbf{x} \in \mathcal{X}, C \in \mathcal{C}} \frac{\pi(\mathbf{x}_C)}{p_C(\mathbf{x}_C)}.$$

The implication of this theorem is that the FGM-equipped function approximator does not require the dataset to cover the entire design space. Rather, it only requires that the individual cliques of the space be covered, which is a much milder requirement, especially when the cliques are small. In the next section, we show how these results can be combined with a transformer, mitigate the distribution shift problem, and enable efficient MBO.

## 3 CLIQUEFORMER

This section introduces a neural network model designed to solve MBO problems through standard end-to-end training on offline datasets. We present a new theoretical result, outline the key desiderata for such a model, and propose an architecture—*Cliqueformer*—that addresses these requirements.

### 3.1 NON-UNIQUENESS OF STRUCTURE DISCOVERY

The regret bound from Theorem 1 applies to methods that use the target function's FGM in their function approximation. It implies that such methods can solve even very high-dimensional problems if their underlying FGMs have low-dimensional cliques or, simply speaking, are *sparse*. Since, in general, no assumptions about the input can be made, this motivates learning a representation of the input for which one can make distributional assumptions and infer the FGM with statistical tests. Following this reasoning, Grudzien et al. (2024) offer a heuristic technique for discovering an FGM over learned, latent, normally-distributed variables. However, as we formalize with the following theorem, even such attempts are futile in dealing with black-box functions.

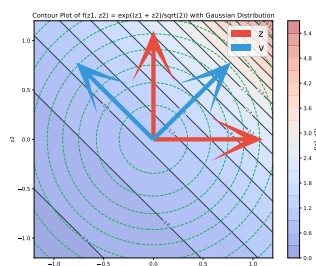

**Theorem 2.** *Let $d \geq 2$ be an integer and $\mathbf{x} \in \mathbb{R}^d$ be a random variable with positive density in $\mathbb{R}^d$. There exists a function $f(\mathbf{x})$ and two different reparameterizations, $\mathbf{z} = z(\mathbf{x})$ and $\mathbf{v} = v(\mathbf{x})$, of $\mathbf{x}$, that both follow a standard-normal distribution, but the FGM of $f$ with respect to $\mathbf{z}$ is a complete graph (has all possible edges), and with respect to $\mathbf{v}$ it is an empty graph (has no edges).*

*Proof Sketch. The proof is a construction. We first map $\mathbf{x}$ to a standard-normal variable $\mathbf{z}$ with tools from high-dimensional statistics. We then introduce scalar variable $y$ that is a function of $\mathbf{z}$ and has a complete FGM. We then show that we can rotate $\mathbf{z}$ onto another normal variable $\mathbf{v}$ with respect to which $y$ is a function with an empty FGM (see Figure 3 for illustration). We complete the proof by showing how to express $y$ as a function of $\mathbf{x}$. The full proof can be found in Appendix A.*

Figure 3: Illustration of construction in the proof of Theorem 2 for $d = 2$. Red axes represent $\mathbf{z}$ and blue axes represent $\mathbf{v}$. When considered a function of $\mathbf{z}$, the contour curves (straight lines) of $f$ are a function of both $z_1$ and $z_2$, but as a function of $\mathbf{v}$, they only depend on $v_1$. The density of the Gaussian distribution (green curves) are identical circles for both variables.

The theorem implies that FGM is not a fixed attribute of a function that can be estimated from the data, but instead should be viewed as a property of the input's reparameterization. Furthermore, different reparameterizations feature different FGMs with varied levels of decomposability, some of which may not significantly simplify the target function. This motivates a reverse approach that starts by defining a desired FGM and learning representations of the input that align with the graph. In the next subsection, we introduce Cliqueformer, where the FGM is specified as a hyperparameter of the model and a representation of the data that follows its structure is learned.

### 3.2 ARCHITECTURE

The goal of this subsection is to derive an MBO model that can simultaneously learn the target function as well as its structure, and thus be readily applied to MBO. First, we would like the model to decompose its prediction into a sum of models defined over small subsets of the input variables in the manner of Equation (3). As discussed in the previous subsection, efforts to discover such a structure are impractical since there exists a plethora of reparameterizations of the data and their

corresponding FGMs of varied decomposabilities. Thus, instead, we propose that the FGM be defined first, and a representation of the data be learned to align with the chosen structure.

**Desideratum 1.** *The model should follow a pre-defined FGM decomposition in a learned space.*

Accomplishing this desideratum is simple. A model that we train should, after some transformations, split the input's representations, denoted as $\mathbf{z}$, into partially-overlapping cliques, $(\mathbf{z}_{C_1}, \ldots, \mathbf{z}_{C_{N_{\text{clique}}}})$, and process them independently, followed by a summation. To implement this, we specify how many cliques we want to decompose the function into, $N_{\text{clique}}$, as well as their dimensionality, $d_{\text{clique}}$, and the size of their *knots*—dimensions at which two consecutive cliques overlap, $d_{\text{knot}} = |\mathbf{z}_{C_i} \cap \mathbf{z}_{C_{i+1}}|$. Then, we pass each clique through an MLP network that is also equipped with a trigonometric clique embedding, similar to Vaswani et al. (2017) to express a different function for each clique,

$$f_\theta(\mathbf{z}) = \frac{1}{N_{\text{clique}}} \sum_{i=1}^{N_{\text{clique}}} f_\theta(\mathbf{z}_{C_i}, \boldsymbol{c}_i), \text{ where } \begin{bmatrix} \boldsymbol{c}_{i,2j} & \boldsymbol{c}_{i,2j+1} \end{bmatrix} = \begin{bmatrix} \sin(i \cdot \omega_j) & \cos(i \cdot \omega_j) \end{bmatrix},$$

and $\omega_j = 10^{-8j/d_{\text{model}}}$. Here, we use the arithmetic mean over cliques rather than summation because, while being functionally equivalent, it provides more stability when $N_{\text{clique}} \to \infty$. Pre-defining the cliques over the representations allows us to avoid the problem of discovering arbitrarily dense graphs, as explained in Subsection 3.1. This architectural choice implies that the regret from Theorem 1 will depend on the coverage of such representations' cliques, $\max_{i \in [N_{\text{clique}}]} 1/e_\theta(\mathbf{z}_{C_i})$, where $e_\theta(\mathbf{z}) = \mathbb{E}_{\mathbf{x} \sim \mathcal{D}}[e_\theta(\mathbf{z}|\mathbf{x})]$ is the marginal distribution of the representations learned by an encoder $e_\theta$. This term can become dangerously large if individual distributions $e_\theta(\mathbf{z}_C)$ put disproportionally more density to some regions of the latent space than to others, Thus, to prevent that, we propose to train the latent space so that the distribution of cliques attain wide coverage.

**Desideratum 2.** *The model should learn representations whose cliques have dsirtibutions featured by wide coverage.*

To meet this requirement, we leverage tools from representation learning, but in a novel way. Namely, we put a variational bottleneck (Kingma & Welling, 2013; Higgins et al., 2016; Alemi et al., 2016, VIB) on **individual** cliques of our representations that brings their distribution closer to a prior with wide coverage, which we choose to be the standard-normal prior. To implement it, when computing the loss for a single example, we sample a single clique to compute the VIB for at random, as opposed to computing it for the joint latent variable like in the classical VIB,

$$\text{VIB}(\mathbf{x}, \theta) = \mathbb{E}_{i \sim U[N_{\text{clique}}]} \big[ \text{KL}\big(e_\theta(\mathbf{z}_{C_i}|\mathbf{x}), p_{C_i}(\mathbf{z}_{C_i})\big) \big], \tag{4}$$

where $e_\theta(\mathbf{z}_C|\mathbf{x})$ is the density of clique $C$ produced by our learnable encoder $e_\theta(\mathbf{z}|\mathbf{x})$, and $p_C(\mathbf{z}_C)$ is the density of $\mathbf{z}_C$ under the standard-normal distriuion. Note that it is not equivalent to the classical, down-weighted VIB in expectation either since our cliques overlap, meaning that knots contribute to the VIB more often than regular dimensions. Together with the model $f_\theta$ and the encoder $e_\theta$, we train a decoder $d_\theta(\mathbf{x}|\mathbf{z})$ that reconstructs the designs from the latent variables. Putting it all together, the training objective of our model is a VIB-style likelihood objective with a regression term,

$$L_{\text{clique}}(\theta) = \mathbb{E}_{(\mathbf{x},\mathbf{y}) \sim \mathcal{D}, \mathbf{z} \sim e_\theta(\cdot|\mathbf{z}), i \sim U[N_{\text{clique}}]} \Big[ \text{VIB}(\mathbf{x}, \theta) - \log d_\theta(\mathbf{x}|\mathbf{z}) + \tau \cdot \big(\mathbf{y} - f_\theta(\mathbf{z})\big)^2 \Big], \tag{5}$$

where $\tau$ is a positive coefficient that we set to 10 in our experiments.

Since in our neural network we impose the FGM decomposition of the predictive module in the latent space, we need to endow our model with high expressivity to learn representations that meet such demands. Thus, we model our encoder $e_\theta$ and decoder $d_\theta$ with transformer networks (Vaswani et al., 2017). To leverage such an architecture, the encoder begins by mapping the input vector $\mathbf{x} \in \mathbb{R}^d$ into $d$ vectors of dimensionality $d_{\text{model}}$, which it then processes as if they were a sequence of token embeddings. After a series of transformer blocks, the network then maps the sequence into a normal distribution over representations, $e_\theta(\mathbf{z}|\mathbf{x})$. A vector $\mathbf{z}$ can then be sampled from that distribution and arranged into $N_{\text{clique}}$ cliques with dimensionality $d_{\text{clique}}$ and knot size of $d_{\text{knot}}$. The new sequence can then be fed into the predictive model $f_\theta$ and to the decoder $d_\theta$, which is a transformer too. For illustration of the information flow in Cliqueformer's training consult Figure 4.

### 3.3 Optimizing Designs With Cliqueformer

Once Cliqueformer is trained, we use it to optimize new designs. Typically, MBO methods initialize this step at a sample of designs $(\mathbf{x}^{i_b})_{b=1}^B$ drawn from the dataset (Trabucco et al., 2021). Since in

Figure 4: Illustration of information flow in Cliqueformer's training. Data are shown in navy, learnable variables in blue, neural modules in pink, and loss functions in green. The input $\mathbf{x}$ is passed to a transformer encoder to compute representation $\mathbf{z}$ which is decomposed into cliques with small overlapping knots (highlighted in colors on the figure). The representation goes to the parallel MLPs whose outputs, added together, predict target y. The representation $\mathbf{z}$ is also fed to a transformer decoder that tries to recover the original input $\mathbf{x}$. Additionally, the representation goes through an information bottleneck from Equation (4) during training.

---

**Algorithm 1** MBO with Cliquefrormer

1: Initialize the encoder, decoder, and predictive model $(e_\theta, d_\theta, f_\theta)$.
2: **for** $t = 1, \ldots, T_{\text{model}}$ **do**
3:   Take a gradient step on the parameter $\theta$ with respect to $L_{\text{clique}}(\theta)$ from Equation (5).
4: **end for**
5: Sample $B$ examples $\mathbf{x}^{(i_b)} \sim \mathcal{D}$, $b \in [B]$, from the dataset.
6: Encode the examples with the encoder $\mathbf{z}^{(i_b)} \sim e_\theta(\mathbf{z}|\mathbf{x}^{(i_b)})$, for $b \in [B]$.
7: **for** $t = 1, \ldots, T_{\text{design}}$ **do**
8:   Decay the representation $\mathbf{z}$ of the design, $\mathbf{z}^{(i_b)} \leftarrow (1-\lambda)\mathbf{z}^{(i_b)}$, $\forall b \in [B]$.
9:   Take a gradient ascent step on the parameter $\mathbf{z}$ with respect to $\hat{\eta}(\mathbf{z})$ from Equation (6).
10: **end for**
11: Propose solution candidates by decoding the representations, $\mathbf{x}^\star \sim d_\theta(\mathbf{x}|\mathbf{z}^\star)$.

---

our algorithm the optimization takes place in the latent space $\mathcal{Z}$, we perform this step by encoding the sample of designs with Cliqueformer's encoder, $\mathbf{z}^{i_b} \sim e_\theta(\mathbf{z}|\mathbf{x}^{i_b})$. We then optimize the representation $\mathbf{z}^{i_b}$ of design $\mathbf{x}^{i_b}$ to maximize our model's value,

$$L_{\text{mbo}}\big((\mathbf{z}^{i_b})_{b=1}^B\big) = \frac{1}{B}\sum_{b=1}^B f_\theta(\mathbf{z}^{i_b}), \tag{6}$$

at the same time minding the enumerator of the regret bound from Theorem 1. That is, we don't want the optimizer to explore regions under which the marginal densities $e_\theta(\mathbf{z}_C) = \mathbb{E}_{\mathbf{x}\sim\mathcal{D}}[e_\theta(\mathbf{z}_C|\mathbf{x})]$ are small. Fortunately, since the encoder was trained with standard-normal prior on the cliques, $p(\mathbf{z}_C) = N(0_{d_{\text{clique}}}, I_{d_{\text{clique}}})$, we know that values of $\mathbf{z}$ closer to the origin have unilaterally higher marginals. This simple property of standard-normal distribution allows us to confine the optimizer's exploration to designs with in-distribution cliques by exponentially decaying the design at every optimization step. Thus, we use AdamW as our optimizer (Loshchilov et al., 2017). We provide the pseudocode of the whole procedure of designnign with Cliqueformer in Algorithm 1.

## 4 RELATED WORK

The idea of using machine learning models in optimization problems has existed for a long time, and has been mainly cultivated in the literature on Bayesian optimization (Williams & Rasmussen, 2006; Brochu et al., 2010; Snoek et al., 2012, BO). The BO paradigm relies on two core assumptions: availability of data of examples paired with their target function values, as well as access to an oracle that allows a learning algorithm to query values of proposed examples. Thus, similarly to

reinforcement learning, the challenge of BO is to balance exploitation and exploration of the black-box function modeled by a Gaussian process. Recently, to help BO tackle very high-dimensional problems, techniques of decomposing the target function have become more popular (Kandasamy et al., 2015; Rolland et al., 2018). Most commonly, these methods decompose the target function into functions defined on the input's partitions. While such models are likely to deviate far from the functions' ground-truth structure, one can derive theoretical guarantees for a range of such decompositions under the BO's query budget assumptions (Ziomek & Bou-Ammar, 2023). However, these results do not apply to our setting of offline MBO, where no additional queries are available and the immediate reliability on the model is essential. Furthermore, instead of partitioning the input, we decompose our prediction over a latent variable that is learned by a transformer, enabling the model to acquire an expressive structure over which the decomposition is valid.

Offline model-based optimization (MBO) has been recently attracting attention of researchers and practitioners from domains where BO assumptions cannot be easily met, offering an attractive premise of producing solutions directly after training on static datasets, without the need for additional queries. One of the first tasks tackled by MBO was molecule design (Gómez-Bombarelli et al., 2018), where a variational auto-encoder (Kingma & Welling, 2013, VAE) was used to learn continuous representations of molecular data. In contrast to our work, however, this work does not study learning structural properties of the target function, but rather is a proof of concept of applying deep learning to molecular design. A data type-agnostic algorithm was introduced by Brookes et al. (2019), dubbed Conditioning by Adaptive Sampling (CBaS), that iteratively refines its design proposals in response to predictions of a non-differentiable oracle. While one can use this refinement procedure in combination with trainable models by means of *auto-focusing* (Fannjiang & Listgarten, 2020), this setting is different than ours since we assume the ability to model the black-box function with a neural network. Since such a model is differentiable, we can simply rely on automatic differentiation (Paszke et al., 2019) to refine our designs. Trabucco et al. (2021) introduced a neural network-based method, exactly for our setting, dubbed Conservative Objective Models (COMs), where a surrogate model is trained to both predict values of examples that can be found in the dataset, and penalize those that are not. COMs differs from our work fundamentally, since its contribution lies in the formulation of the conservative regularizer applied to arbitrary neural networks, while we focus on scalable model architectures that facilitate computational design. Another recent line of work proposes to tackle the design problem through means of generative modeling. BONET (Mashkaria et al., 2023) and DDOM (Krishnamoorthy et al., 2023) are examples of works that bring the most recent novelties of the field to address design tasks. BONET does so by training a transformer to generate sequences of designs that monotonically improve in their value, and DDOM by training a value-conditioned diffusion model. That is, these methods attempt to generate high-value designs through novel conditional generation mechanisms. Instead, we model MBO as a maximization problem, and propose a scalable model that acquires the structure of the black-box function through standard gradient-based learning. To this end, we bring powerful techniques from deep learning and generative modeling, like transformers (Vaswani et al., 2017) and variational-information bottlenecks (Kingma & Welling, 2013; Alemi et al., 2016).

The work of Grudzien et al. (2024) introduced the theoretical foundations of functional graphical models (FGMs), including Theorem 1. However, as we have shown in Theorem 2, their graph discovery heuristic for neural networks renders learning the black-box function's structure an open problem. Our work addresses this issue by subsuming the graph discovery step in the architecture of our *Cliqueformer* that learns to abide by a pre-defined FGM. As such, the model learns the structure of the target function, as well as learns to predict its value, in synergy within an end-to-end training.[2]

## 5 Experiments

In this section, we provide the empirical evaluation of Cliqueformer. We begin by benchmarking Cliqueformer against prior methods on tasks from the MBO literature. Then, we finish by evaluating the benefit of the novel FGM decomposition layer in Cliqueformer through an ablation study.

### 5.1 Benchmarking

We compare our model to three classes of algorithms, each represented by a proven prior method. As a *naïve* baseline, we employ gradient ascent on a learned model (*Grad. Asc.*). To compare to

---

[2]For more related work, please see Appendix D.

| Task | Grad.Asc. | RWR | COMs | DDOM | Transformer | Cliqueformer |
|---|---|---|---|---|---|---|
| LRBF 11 | $-\infty \pm 0.00$ | $0.08 \pm 0.08$ | $\mathbf{0.66} \pm 0.04$ | $-\infty \pm 0.00$ | $0.47 \pm 0.05$ | $0.65 \pm 0.07$ |
|  | 0% | 1% | 72% | 0% | 60% | **74%** |
| LRBF 31 | $-\infty \pm 0.00$ | $0.31 \pm 0.10$ | $0.50 \pm 0.05$ | $-\infty \pm 0.00$ | $-\infty \pm 0.00$ | $\mathbf{0.64} \pm 0.05$ |
|  | 0% | 3% | 32% | 0% | 0% | **76%** |
| LRBF 41 | $-\infty \pm 0.00$ | $0.35 \pm 0.08$ | $0.45 \pm 0.06$ | $-\infty \pm 0.00$ | $0.20 \pm 0.01$ | $\mathbf{0.66} \pm 0.05$ |
|  | 0% | 3% | 16% | 0% | **75%** | 72% |
| LRBF 61 | $-\infty \pm 0.00$ | $0.29 \pm 0.10$ | $0.25 \pm 0.04$ | $-\infty \pm 0.00$ | $0.16 \pm 0.03$ | $\mathbf{0.66} \pm 0.05$ |
|  | 0% | 4% | 7% | 0% | 64% | **68%** |
| Superconductor | $1.13 \pm 0.08$ | $1.03 \pm 0.07$ | $0.97 \pm 0.08$ | $1.22 \pm 0.08$ | $0.96 \pm 0.05$ | $\mathbf{1.43} \pm 0.04$ |
| TF-Bind-8 | $0.99 \pm 0.00$ | $\mathbf{1.58} \pm 0.03$ | $1.57 \pm 0.02$ | $1.55 \pm 0.03$ | $1.48 \pm 0.03$ | $\mathbf{1.58} \pm 0.01$ |
| DNA HEPG2 | $\mathbf{2.16} \pm 0.07$ | $1.91 \pm 0.12$ | $1.20 \pm 0.09$ | $1.82 \pm 0.10$ | $2.13 \pm 0.06$ | $2.10 \pm 0.07$ |
| DNA k562 | $2.11 \pm 0.06$ | $1.91 \pm 0.11$ | $1.80 \pm 0.12$ | $2.61 \pm 0.21$ | $2.60 \pm 0.19$ | $\mathbf{3.15} \pm 0.07$ |
| Ave.score $\uparrow$ | 0.80 | 0.93 | 0.93 | 0.90 | 1.00 | **1.36** |
| Ave.rank $\downarrow$ | 4.75 | 3.88 | 3.63 | 4.50 | 4.25 | **1.38** |

Table 1: Experimental results of Cliqueformer and the baselines. Each score is the mean of values of $TopK$=10 of $B$=1000 designs, averaged over 5 runs. The values were normalized with the min-max scheme, where the minimum and the maximum are taken from the dataset, so that the scores of the designs in the dataset are in range $[0, 1]$. We note that, unlike Trabucco et al. (2022), we take the maximum from the data available for the MBO model training (the union of the train and test data), and not from the oracle training data, to make the results more interpretable (see Appendix C). We also provide the standard deviation estimates as the standard deviation for each of the $TopK$ samples, averaged over the runs. Additionally, for LRBF tasks, we provide the average validity in blue, calculated as the percentage of valid designs from the $B$ produced candidates, averaged across the runs. We provide average score (the higher the better) and the average rank for each method. For the average score $-\infty$ was taken into calculation as zero.

*exploratory* methods, we use Reward-Weighted Regression (Peters & Schaal, 2007, *RWR*) which learns by regressing the policy against its most promising perturbations. To represent the recently proposed *conservative* algorithms, we compare to state-of-the-art Conservative Objective Models (Trabucco et al., 2021; Kumar et al., 2021, *COMs*). For COMs, we use the recommended hyper-parameter setting from (Trabucco et al., 2021). While Cliqueformer can be tuned for each specific task, we keep most of the hyperparameters the same: refer to Appendix C for more details.

For every method, we report an empirical estimate of its $100^{th}$ percentile, similarly to Trabucco et al. (2021). We estimate it by averaging the values of the top 10 designs out of 1000 candidates, averaged across 5 seeds. In the following paragraphs, we introduce benchmark tasks, from MBO literature, that we use in our experiments.

**Latent Radial-Basis Functions (LRBF).** This is a suite of tasks designed to expose vulnerability of MBO models (Grudzien et al., 2024). The data pairs $(\mathbf{x}, \mathbf{y})$ are generated by first drawing a standard normal vector $\mathbf{z} \sim N(0_{d_z}, I_{d_z})$, then computing y as a sum of radial-basis functions of $d_C$-dimensional cliques of a pre-defined FGM. The observed vector $\mathbf{x}$ is a non-linear transformation of $\mathbf{z}$, *i.e.*, $\mathbf{x} = T(\mathbf{z}) \in \mathcal{T} \subset \mathbb{R}^d$, where $d > d_z$, while $\mathbf{z}$ itself is hidden from the data. Such tasks allow us to study whether an MBO method learns to produce *valid* designs by verifying that *ground-truth* inputs $\mathbf{z}$ can be recovered from them. That is, a design $\hat{\mathbf{x}}$ for which the map $T^{-1}(\hat{\mathbf{x}})$ is ill-defined is considered invalid. In our experiments, an invalid design receives a value of $-\infty$. We report the average validity of designs produced by the methods in blue. The results in Table 1 show that the most able method at keeping its proposals at the manifold of valid designs is Cliqueformer. Importantly, this ability does not diminish even in higher-dimensional tasks, while the second-best such method, COMs, gradually loses this ability. We also use these tasks to study if a model is capable of exploiting the RBF's structure in the optimization step by varying the effective dimensionality $d_z$ of the data while keeping $d_C$ fixed. While the naive and the exploratory baseline perform poorly on these tasks overall, COMs's performance clearly drops as the task dimension increases. Meanwhile, as predicted by Theorem 1, Cliqueformer attains similar, strong performance across all tasks.

**Superconductor.** This task poses a challenge of designing a superconducting material, represented by an 81-dimensional vector, with as high critical temperature as possible (Hamidieh, 2018). It tests

abilities of MBO models in real-world continuous problems. As the very high score of gradient ascent shows, in contrast to LRBF, this task rewards greedy optimizer updates. While we developed Cliqueformer paying attention to distribution shift, its ability to compose individual in-distribution cliques into a joint solution (*stitching*) allows for large improvements that outperform all baselines.

**TFBind-8 & DNA Enhancers.** In these discrete tasks we optimize DNA sequences of length 8 and 200, respectively. In TFBind-8 the target is the sequence's binding affinity with a particular transcription factor, while in DNA Enhancers we maximize HEPG2 and k562 activity levels. Being low-dimensional, TFBind-8 (Trabucco et al., 2022) is a testbed that allows us to verify MBO models' ability to solve discrete tasks. Cliqueformer solves this problem very efficiently, largely improving upon the dataset, but it is worth noting that most baselines, with the exception of gradient ascent, performed similarly. Thus, while the TFBind-8 task does not favor greedy design optimization, Cliqueformer is still able to leverage its other strengths to achieve great performance. Then, we use the DNA Enhancers tasks (Uehara et al., 2024) to study the scalability of our method to very high-dimensional problems and large datasets (approximately $2 \times 10^5$ examples in this case). The results in Table 1 and the high score of gradient ascent show that these tasks favor direct, greedy optimization of designs more than they benefit from conservatism of COMs. Nevertheless, Cliqueformer is able to greatly exceed the quality of observed designs, and performs on par with gradient ascent in HEPG2, and greatly outperforms all baselines in k562, confirming its ability to learn structure within discrete and very high-dimensional data. Overall, averaging across all tested tasks, Cliqueformer achieves the best overall performance.

## 5.2 ABLATIONS

While the decomposing elements of Cliqueformer are novel, other components, such as transformer blocks (Vaswani et al., 2017) and variational information bottlenecks (Kingma & Welling, 2013; Alemi et al., 2016), are components proposed in prior work (albeit for a different purpose). In this subsection, we verify the utility of the decomposing component with an ablation study, in which we sweep over the number of cliques of Cliqueformer for a few representative tasks. In each of the tested tasks we fix the size of the latent variable $\mathbf{z}$ and sweep over the number of cliques $N_{\text{clique}}$ into which it can be decomposed. We cover the case $N_{\text{clique}} = 1$ to compare Cliqueformer to an FGM-oblivious VAE with transformer backbone and AdamW design optimizer.

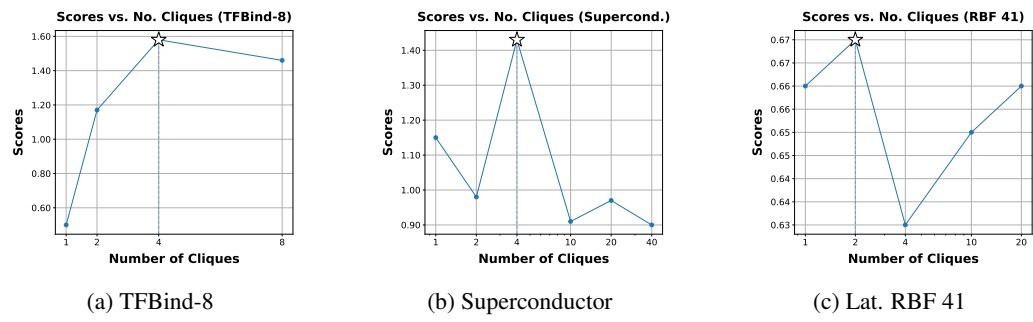

| (a) TFBind-8 | (b) Superconductor | (c) Lat. RBF 41 |

Figure 5: Ablation experiments on the number of cliques used in the FGM decomposition of Cliqueformer. For each task, we fix the size of the latent variable and the overlap size, and sweep over possible $N_{\text{clique}}$ values. We use TFBind-8 (left), Superconductor (center), and Lat. RBF 41 (right) tasks. The x-ais (log-scale) denotes the number of cliques, and the y-axis is the final score of the model.

Results in Figure 5 show that Cliqueformer does benefit from the FGM decomposition in all tested tasks. However, the results suggest that the optimal number of cliques varies between tasks – *e.g.*, 4 for TFBind-8 and Superconductor, but 2 for LRBF 41. In our experiments, we consistently obtained good performance by setting the clique size to $d_{\text{clique}} = 3$ and choosing the number of cliques so that the total latent dimension approximately matches that of the design. For DNA Enhancers, we doubled the clique size and halved the number of cliques to decrease the computational cost of attention. More details of hyperparameters can be found in Appendix C. An in-depth analysis of the relation between hyperparameters and the performance is an exciting avenue of future work.

## 6    CONCLUSION

In this work, we proposed Cliqueformer, a scalable architecture for model-based optimization. We derived its building blocks following recent advances in MBO theory centered in functional graphical models, equipping it with an ability to acquire structure of the black-box target function. This ability sets the model free from requiring explicit conservative regularization or iterative retraining to propose in-distribution designs. Empirically, Cliqueformer outperforms all baselines across all tested tasks. Cliqueformer opens an exciting avenue of research in MBO focused on scaling design datasets and large neural networks.

### REPRODUCIBILITY STATEMENT

We provide our code base, implemented in Pytorch, in the supplementary material. It contains files with default hyper-parameters for both Cliqueformer and the baselines. We also set default random seeds in the training script (`training.py`) that can be used to reproduce some of our runs exactly. This script saves a pre-trained model that one can use to optimize designs with by running `optimize.py`. To enable reproducing Cliqueformer's performance, we provide a table with hyper-parameters in Appendix C. We ran most of our experiments on a machine with an Nvidia Titan X GPU, with the exception of DNA Enhancers tasks which we ran on a Google TPU v3-8.

Due to lack of the off-the-shelf availability of Design Bench, we scraped the data of the benchmark tasks from prior works' repositories. We adopted the implementation of LRBF tasks from Grudzien et al. (2024)'s code at

```
https://colab.research.google.com/drive/
1qt4M3C35bvjRHPIpBxE3zPc5zvX6AAU4?usp=sharing
```

We used Superconductor data from Fannjiang & Listgarten (2020)'s code on

```
https://github.com/clarafy/autofocused-oracles.
```

Following the authors, we train a boosted tree model to serve as an oracle for new designs. We used TFBind-8 from

```
https://huggingface.co/datasets/beckhamc/design_bench_data/tree/
main/tf_bind_8-SIX6_REF_R1,
```

which comes with values for all possible designs that can be looked up at evaluation. We obtained the DNA Enhancers data from the recent tutorial on offline fine-tuning of generative models at

```
https://github.com/masa-ue/RLfinetuning_Diffusion_Bioseq/tree/
master/tutorials/Human-enhancer,
```

whose pre-trained oracle we used for evaluation.

We will release publically our code on github upon the paper's publication.

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

APPENDIX

## A   THEORETICAL DETAILS

The full statement of the following theorem considers an MBO algorithm with function clas $\mathcal{F} = \mathcal{F}_{C_1} \oplus ... \oplus \mathcal{F}_{C_{N_{\text{clique}}}}$, so that every of its element has form

$$f(\mathbf{x}) = \sum_{i=1}^{N_{\text{clique}}} f_{C_i}(\mathbf{x}_{C_i}).$$

As described in Section 3, Cliqueformer's architecture forms such a function class on top of the learned latent space. We define the statistical constant as

$$C_{\text{stat}} = \sqrt{\frac{1}{1-\sigma}}, \quad \text{where } \sigma = \max_{C_i \neq C_j, \hat{f}_{C_i}, \hat{f}_{C_j}} \mathbb{C}orr_{\mathbf{x} \sim p}[\hat{f}_{C_i}(\mathbf{x}_{C_i}), \hat{f}_{C_j}]$$

and the function approximation complexity constant as

$$C_{\text{cpx}} = \sqrt{\frac{N_{\text{clique}} \sum_{i=1}^{N_{\text{clique}}} \log(|\mathcal{F}_{C_i}|/\delta)}{N}},$$

where $\delta$ is the PAC error probability (Shalev-Shwartz & Ben-David, 2014).

**Theorem 1** (Grudzien et al. (2024)). *Let $f(\mathbf{x})$ be a real-valued function, $\mathcal{C}$ be the set of maximal cliques of its FGM, and $\Pi$ be a policy class. Let $C_{stat}$ and $C_{cpx}$ be constants that depend on the probability distribution of $\mathbf{x}$ and function approximator class's complexity, respectively, defined in Appendix A. Then, the regret of MBO with the FGM information is given by,*

$$\eta(\pi^\star) - \eta(\hat{\pi}_{FGM}) \leq C_{stat} C_{cpx} \max_{\pi \in \Pi, \mathbf{x} \in \mathcal{X}, C \in \mathcal{C}} \frac{\pi(\mathbf{x}_C)}{p_C(\mathbf{x}_C)}.$$

**Theorem 2.** *Let $d \geq 2$ be an integer and $\mathbf{x} \in \mathbb{R}^d$ be a random variable with positive density in $\mathbb{R}^d$. There exists a function $f(\mathbf{x})$ and two different reparameterizations, $\mathbf{z} = z(\mathbf{x})$ and $\mathbf{v} = v(\mathbf{x})$, of $\mathbf{x}$, that both follow a standard-normal distribution, but the FGM of $f$ with respect to $\mathbf{z}$ is a complete graph (has all possible edges), and with respect to $\mathbf{v}$ it is an empty graph (has no edges).*

*Proof.* Since the density of $\mathbf{x}$ is positive and continuous, we can form a bijection that maps $\mathbf{x}$ to another random variable $\mathbf{z} \in \mathbb{R}^l$, where $l \leq d$, that follows the standard-normal distribution (Dai & Wipf, 2019, Appendix E). We denote this bijection as $Z(\mathbf{x})$. Let us define

$$\mathbf{y} = f^z(\mathbf{z}) = \exp\left(\frac{1}{\sqrt{l}} \sum_{i=1}^{l} \mathbf{z}_i\right).$$

Then, the FGM of $f^z$ has an edge between every two variables since each variable's partial derivative

$$\frac{\partial f^z}{\partial \mathbf{z}^i} = \frac{1}{\sqrt{l}} \exp\left(\frac{1}{\sqrt{l}} \sum_{i=1}^{d} \mathbf{z}_i\right)$$

is also a function of all others (Grudzien et al., 2024, Lemma 1). Consider now a rotation $\rho : \mathbf{z} \mapsto \mathbf{v} = (\mathbf{v}_1, \ldots, \mathbf{v}_l)$ such that $\mathbf{v}_1 = \frac{1}{\sqrt{l}} \sum_{i=1}^{l} \mathbf{z}_i$. Then, $\mathbf{v} \sim N(0_l, I_l)$, and $\mathbf{y}$ can be expressed in terms of $\mathbf{v}$ as $\mathbf{y} = f^v(\mathbf{v}) = \exp(\mathbf{v}_1)$. Then, the FGM of $f^v$ has no edges, since it depends on only one variable, inducing no interactions between any two variables. Recall that $\mathbf{x} = Z^{-1}(\mathbf{z})$. Then, $\mathbf{x}$ be represented by standard-normal $\mathbf{z}$ and $\mathbf{v}$, obtainable by

$$\mathbf{z} = Z(\mathbf{x}) \text{ and } \mathbf{v} = \rho(\mathbf{z}) = \rho\big(Z(\mathbf{x})\big).$$

Furthermore, we can define

$$f(\mathbf{x}) = f^z\big(Z(\mathbf{x})\big)$$

which is identically equal to $f^z(\mathbf{z})$ and $f^v(\mathbf{v})$, which have a complete and an empty FGM, respectively, thus fulfilling the theorem's claim. $\qquad\square$

## B REPRESENTATION DISTRIBUTION

In this section, we study the distribution of the latent representations $\mathbf{z}$ that were trained with Equation (5). The loss regulates the cliques of the latent, although not the joint variable, to have marginals close to the normal $N(0_{d_{clique}}, I_{d_{clique}})$ distribution. We examine the latents in TFBind-8 and LRBF41, where $d_{clique} = 3$, to see if their distributions display standard normal-like properties (low-magnitude mean and off-diagonal covariances).

Namely, in each task, we take a trained Cliqueformer, sample a batch of 1000 designs $\mathbf{x} \sim \mathcal{D}$ from the dataset, and encode it with the model's encoder, $\mathbf{z} \sim e_\theta(\mathbf{z}|\mathbf{x})$. We then compute the sample mean and the sample covariance matrix. We scatter-plot the mean values against the coordinates, and plot the heat-map of the sample covariance whose diagonal is zeroed-out (we are interested in covariances more than in variances), and whose entries are passed through the absolute value function (we are interested in the magnitude of covariance). Additionally, for LRBF 41, we plot the average-smoothed (with the 11x11 kernel) version of the covariance matrix to suppress the effect of outliers. The results for TFBind-8 can be found in Figure 6, and for LRBF 41 in Figure 7.

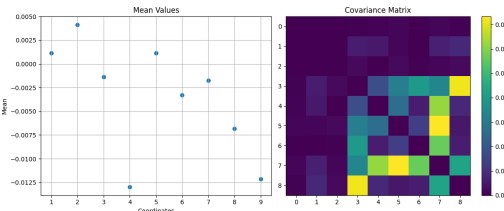

Figure 6: TFBind-8

In TFBind-8, the mean values of all coordinates are within the $[-0.01, 0.01]$ interval, and all covariance values are within $[0, 0.07]$, indicating standard normal-like behavior of the latents, beyond what was required - standard normality of individual cliques.

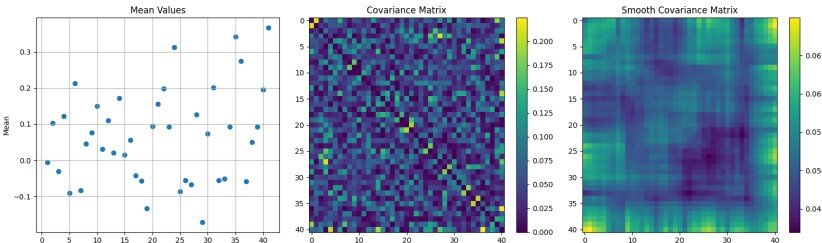

Figure 7: LRBF 41

In the higher-dimensional LRBF 41, the means are low-magnitude too, but the strength of this property is much lower: there are three coordinates whose sample mean exceeds $0.3$. The entries of the covariance matrix are still quite low - up to $0.20$ in rare cases, and they get lowered down to $0.07$ after average-smoothing due to the prevalence of near-zero entries. The smoothing also reveals that the highest covariance, the bright-yellow regions, occur in the lowest row of the smoothed matrix. These entries correspond to covariances between the last coordinate $z_{40}$ and all other coordinates. This is expected because of the clique-based VIB that does not regulate the covariance between $z_{40}$ and other entries often.

## C EXPERIMENTAL DETAILS

**Datasets.** We use the implementation of Grudzien et al. (2024) to generate data with latent radial-basis functions. Also, we initially wanted to use Design Bench (Trabucco et al., 2022) for experiments with practical tasks. However, at the time of this writing, the benchmark suite was suffering

a data loss and was not readily available. To overcome it, we manually found the data and implemented dataset classes. TFBind-8 (Trabucco et al., 2022) could be fully downloaded since the number of possible pairs $(\mathbf{x}, \mathbf{y})$ is quite small. Hence, a design can be evaluated by looking up its score in the dataset. For Superconductor (Hamidieh, 2018), we pre-trained an XGBoost oracle on the full dataset, and trained our model and the baselines to predict the labels produced by the oracle. The proposed designs of the tested models are evaluated by calling the oracle as well. We obtained DNA Enhancers dataset from the code of Uehara et al. (2024), available at

```
https://github.com/masa-ue/RLfinetuning_Diffusion_Bioseq/tree/master.
```

Following the procedure in

```
https://github.com/masa-ue/RLfinetuning_Diffusion_Bioseq/blob/master/
            tutorials/Human-enhancer/1-Enhancer_data.ipynb.
```

we additionally filter the dataset to keep only sequences featured by chromosomes from 1 to 4. We use their pre-trained oracle for generating labels and evaluation of proposed designs. Following Fannjiang & Listgarten (2020) and Trabucco et al. (2022), we train our models on the portions of the datasets with values below their corresponding $80^{th}$. Upon evaluation, we obtain the ground-truth/oracle value of the proposed design y, and normalize it as

$$\bar{\mathbf{y}} = \frac{\mathbf{y} - \mathbf{y}_{\min}}{\mathbf{y}_{\max} - \mathbf{y}_{\min}},$$

and report $\bar{\mathbf{y}}$. $\mathbf{y}_{\min}$ and $\mathbf{y}_{\max}$ are the minimum and the maximum of the training data. This normalization scheme is different than, for example, the one in the work by Trabucco et al. (2022). We choose this scheme due to its easy interpretability—a score of $\bar{\mathbf{y}} > 1$ implies improvement over the given dataset, which is the ultimate objective of MBO methods. However, we note that a score of less than 1 does not imply failute of the algorithm, since we initialize our designs at a random sample from the dataset, which can be arbitrarily low-value or far from the optimum. For some functions, like in latent RBFs, the optima are very narrow spikes in a very high-dimensional space, being nearly impossible to find (see Figure 1a). We choose such an evaluation scheme due to its robustness that allows us to see how good ut improving any design our algorithms are overall.

**Hyper-parameters.** For baselines, we use hyper-parameters suggested by Trabucco et al. (2021). We decreased the hidden layer sizes (at no harm to performance) for LRBF 31 and DNA Enhancers tasks where the performance was unstable with larger sizes. Also, we haven't tuned most of the Cliqueformer's hyper-parameters *per-task*. We found, however, as set of hyper-parameters that works reasonably well on all tasks.

On all tasks, we use **2 transformer blocks** in both the encoder and the decoder, with **transformer dimension of 64**, and **2-head attention**. The predictive model $f_\theta(\mathbf{z})$ is a multi-layer perceptron with **2 hidden layers of dimension 256**. We change it to 512 only for DNA Enhancers. The best activation function we tested was **GELU** (Hendrycks & Gimpel, 2016), and LeakyReLU(0.3) gives similar results. We use **dropout of rate 0.5** (Srivastava et al., 2014). In all tasks, **weight of the MSE term to $\tau =$10** (recall Equation (5)). Additionally, we warm up our VIB term linearly for 1000 steps (with maximal coefficient of 1). We train the model with AdamW (Loshchilov et al., 2017) with the default weight decay of Pytorch (Paszke et al., 2019). We set the **model learning rate to 1e-4** and the **design learning rate to 3e-4** in all tasks. We train the design with AdamW with high rates of weight decay (ranging from 0.1 to 0.5).

In all tasks, we wanted to keep the dimension of the latent variable $\mathbf{z}$ more-less similar to the dimension of the input variable $\mathbf{x}$, and would decrease it, if possible without harming performance, to limit the computational cost of the experiments. The dimension of $\mathbf{z}$ can be calculated from the clique and knot sizes as

$$dim(\mathbf{z}) = d_{\text{knot}} + N_{\text{clique}} \cdot (d_{\text{clique}} - d_{\text{knot}}).$$

In most tasks, we used the clique dimension $d_{\text{clique}} = 3$ with knot size of $d_{\text{knot}} = 1$. We made an exception for Superconductor, where we found a great improvement by setting $d_{\text{clique}} = 21$ and $N_{\text{clique}} = 4$ (setting $d_{\text{clique}} = 3$ and $N_{\text{clique}} = 40$ gives score of 0.99); and DNA Enhancers, where we

doubled the clique size (to 6) and halved the number of cliques to (40), to lower the computational cost of attention. In DNA Enhancers tasks, we additionally increased the MLP hidden dimension to 512 due to greater difficulty of modeling high-dimensional tasks. We summarize the task-specific hyper-parameters in Table 2.

We want to note that these hyper-parameters are not optimal per-task. Rather, we chose schemes that work uniformly *well enough* on all tasks. However, each task can benefit from further alteration of hyper-parameters. For example, we observed that LRBF tasks benefit from different numbers of design steps; for LRBF 41, we found the optimal number to be 400; for Superconductor, it seems to be 200. Due to time constraints, we have not exploited scalability of Cliqueformer in DNA Enhancers tasks, but observed pre-training losses to decrease more with increased parameter count and training duration.

| Task | N_clique | d_clique | MLP dim | design steps | Weight decay |
|---|---|---|---|---|---|
| **LRBF 11** | 10 | 3 | 256 | 50 | 0.5 |
| **LRBF 31** | 18 | 3 | 256 | 50 | 0.5 |
| **LRBF 41** | 20 | 3 | 256 | 50 | 0.5 |
| **LRBF 61** | 28 | 3 | 256 | 50 | 0.5 |
| **TFBind-8** | 4 | 3 | 256 | 1000 | 0.5 |
| **Superconductor** | 4 | 21 | 256 | 1000 | 0.5 |
| **Dna Enhancers** | 40 | 6 | 512 | 1000 | 0.1 |

Table 2: Hyper-parameter configuration for different benchmark tasks.

Below, we list the computational complexities, as the order of the number of FLOPs, for each method's training step and design optimization phase, as a function of batch size $B$, number of model layers $L$, model's hidden dimension $H$, number of exploratory perturbations $P$, number of adversarial training sub-steps $A$, number of design optimization steps $T$, and the number of cliques in an FGM-based model $C$. Note that the majority of quadratic terms, such as $H^2$ and $C^2$ do not influence runtime much if parallelized on a GPU/TPU. We print in bold terms, such as $\mathbf{T}$, that contribute to the complexity with sequential operations, thus inevitably affecting the runtime.

| Method | Training step | Design |
|---|---|---|
| Grad Asc. | $\mathcal{O}(BLH^2)$ | $\mathcal{O}(\mathbf{T}LH^2)$ |
| RWR | $\mathcal{O}(BLH^2)$ | $\mathcal{O}(\mathbf{T}PLH^2)$ |
| COMs | $\mathcal{O}((\mathbf{A}+B)LH^2)$ | $\mathcal{O}(\mathbf{T}LH^2)$ |
| DDOM | $\mathcal{O}(BLH^2)$ | $\mathcal{O}(\mathbf{T}LH^2)$ |
| Transformer | $\mathcal{O}(BLHD(H+D))$ | $\mathcal{O}(\mathbf{T}LHD(H+D))$ |
| Cliqueformer | $\mathcal{O}(BLH(D(H+D)+C(H+C)))$ | $\mathcal{O}(\mathbf{T}LH(D(H+D)+C(H+C)))$ |

Table 3: Computational complexities (in terms of FLOPs) of methods from Section 5.

# D  MORE RELATED WORK

Several reinforcement learning (RL) approaches have been explored extensively for biological sequence design. DyNA-PPO (Angermueller et al., 2019) leverages proximal policy optimization (Schulman et al., 2017) with a model-based variant to improve sample efficiency in the low-round setting typical of wet lab experiments. PEX, also resembling the PPO (Schulman et al., 2017) learning style, (Ren et al., 2022) prioritizes local search through directed evolution (Arnold, 1998) while using a specialized architecture for modeling fitness landscapes. FBGAN (Gupta & Zou, 2018) introduces a feedback loop mechanism to optimize synthetic gene sequences using an external analyzer. However, these methods fundamentally rely on active learning and iterative refinement through oracle queries - DyNA-PPO requires simulator fitting on new measurements, PEX conducts proximal exploration, and FBGAN uses feedback loops with an external analyzer. This makes them unsuitable for offline MBO settings where no additional queries are allowed. Furthermore, while

these approaches are specialized for biological sequences, offline MBO aims to tackle a broader class of design problems through static dataset learning.

