# OpenReview forum: "Cliqueformer: Model-Based Optimization With Structured Transformers"
_ICLR.cc/2025/Conference — Submitted to ICLR 2025_

### Official Review · Reviewer_JStj · 2024-11-03

**Soundness:** 2
**Presentation:** 2
**Contribution:** 2
**Rating:** 5
**Confidence:** 3

**Summary:**

The paper studies the problem of offline model based optimization. The author first consider outline shortcomings of naively using the perspective of recently proposed functional graphical models FGM) perspective for model based optimization. Specifically, the authors show that the FGM is not an attribute of a function to be inferrred but depends on the parameterization of the input. The paper thus considers fixing the structure of the FGM (as a hyperparameters). The paper introduces a transformer-based arhcitecture called Cliqueformer and proposes a training scheme. Unlike previous approaches that try to discover FGMs directly, Cliqueformer learns latent representations that align with a predefined FGM structure. The model combines transformer blocks with a decomposed predictive module and a novel form of variational bottleneck applied to individual cliques. The approach achieves strong performance across various tasks, including synthetic functions, chemical design, and genetic sequence optimization.

**Strengths:**

* Cliqueformer is a novel approach to handling distribution shift in MBO by learning structured representations rather than using conservative regularization.
* The training objective is quite interesting, combining useful ideas from VIB with FGMs.
* The paper is generally well written and easy to follow.
* The submission is accompanied by code to reproduce the results with adequate instructions to reproduce the results.

**Weaknesses:**

* The authors acknowledge that optimal clique configurations vary between tasks, but use a general set of values for their experiments but is very much unclear to me how the parameters would be selected for a new problem? To be clear, this is a challenge in all offline MBO algorithms where selecting the hyperparamters is not trivial (since that requires access to online evaluations) but with the proposed approach there are a lot more hyperparameters (e.g. knowing the optimal clique configurations, the transformer parameters, etc) which makes the results less convincing.  More systematic study of hyperparameter sensitivity would be valuable (especially related to the transformer configuration.
* Another significant flaw is the evaluation of the empirical results and discussion of related works. First the paper does a poor job of covering the offline MBO literature. Several recent works are not discussed in the related work (some cited below). The authors also choose just two baselines, which are quite inadequate considering the highly active nature of the field. To be clear, the issue is that the paper claims state-of-the-art performance but fails to compare to the state-of-the-art. More
* Another important aspect missing from the empirical evaluation is a discussion of the computational complexity of the approach. Specifically how does the runtime of using Cliqueformer compare with the runtime of the baselines. Especially since the Cliqueformer uses a whole transformer, whereas (if I understood correctly) the COM baseline is simply a MLP.
* Finally, there are no formal guarantees about learned representations matching desired FGM structure.

Minor:

* This may be considered a minor nitpick but result of Theorem doesn't seem substantial enough to be a theorem. I would suggest calling it a Proposition.
* The approach of learning a structured representation of a function is very common in the causality and graphical models literature. Even though the paper is specifically about MBO, I encourage the authors to add some discussion about this in the related work section.
* L118: methos -> methods
* L120: com -> COM
* L126-7: “This is particularly frustrating since recent work on” -> why frustrating?
* Figure 3: Axis labels and ticks are too small.
* L316: designign -> designing


[1] Kim, M., Berto, F., Ahn, S., & Park, J. (2023). Bootstrapped training of score-conditioned generator for offline design of biological sequences. Advances in Neural Information Processing Systems, 36.

[2] Chen, C. S., Beckham, C., Liu, Z., Liu, X. S., & Pal, C. (2023). Parallel-mentoring for offline model-based optimization. Advances in Neural Information Processing Systems, 36.

[3] Chen, C., Zhang, Y., Liu, X., & Coates, M. (2023, July). Bidirectional learning for offline model-based biological sequence design. In International Conference on Machine Learning (pp. 5351-5366). PMLR.

[4] Yuan, Y., Chen, C. S., Liu, Z., Neiswanger, W., & Liu, X. S. (2023). Importance-aware co-teaching for offline model-based optimization. Advances in Neural Information Processing Systems, 36.

**Questions:**

* How sensitive is the method to the choice of predefined FGM structure? What guidelines would you suggest for selecting appropriate clique sizes and overlaps for new problems?
* How does the computational cost of Cliqueformer compare to baselines?
* It is not clear to me how this approach would be applicable to problems where the design space is a bit more unstructured (e.g. graphs). Do you have any thoughts on that?

---

> ### Author Response · Authors · 2024-11-19
>
> ## Critique
>
> 1. How would you choose hyper-parameters for a new problem? More systematic study of the hyper-parameter configuration would be of value.
>
> > In our experiments, we found the following scheme to work generally well: *(1)* for problems with $dim(x) < 100$: we fix $d_{clique}=3$ and $d_{knot}=1$, and choose $N_{clique}$ so that $dim(z)=N_{clique}\cdot (d_{clique} - d_{knot}) + d_{knot} \approx dim(x)$; *(2)* for problems with $dim(x) \geq 100$, we fix $d_{clique}=6$ and repeat the procedure, to decrease the computational cost of the attention layer, which is quadratic in $N_{clique}$. We reported the results of these scheme in Table 1 with the exception of Superconductor, where a sweep over these values brought a substantial improvement. However, the result for $d_{clique}=3$ (and $N_{clique}=40$) remains in Figure 5 (b). Would the reviewer like us to put it in the table?
>
> > We understand that a systematic study of these hyper-parameters would be helpful. However, this paper introduces a new transformer architecture that is meant to enable optimization of designs with vanilla optimization techniques, like AdamW (without need for CbAS or conditional generation). We thus spent the majority of time describing the novel model, leaving little room for a systematic hyper-parameter study, and thus providing only the esssential ablation. In fact, this is quite normal for works that introduce new models: AlexNet has not ablated the dropout rate or the momentum parameter [1]; Transformer did not analyze the importance of the attention head dimension per parameter count [2] - such a study came years later [3]. We do consider such analysis of our hyper-parameters, both theoretically and empirically, an exciting avenue of future work.
>
> 2. The spectrum of baselines is limited. More methods must be compared against to claim the state-of-the-art performance.
>
> > We agree with the reviewer. We have added results of the more recent SOTA diffusion-based method DDOM [4], as well as of Gradient Ascent with the transformer backbone,
>
> 3. The authors miss the discussion of the computational complexity of Cliqueformer.
>
> > We understandand if the reviewer would like to see computational complexity analysis, but we do not think that lack of it is a unique flaw of our paper. Most deep learning papers do not offer such analysis. Diffusion models [5] were introduced with a U-Net architecture that added an attention layer at at least one level of resolution, but the computational complexity was omitted. COMs [6] performs 50 adversarial steps for every update (see Line 4 in their Algorithm 1) and does not provide that analysis either.
> Cliqueformer relies on the transformer backbone, so it's computational complexity is that of the transformer. We do not think there is anything unusual about it.
>
> 4. The authors do not provide theoretical guarantees for their method.
>
> > While we agree that, ideally, all computational methods had theoretical guarantees. However, modern deep learning - which we rely on - lacks mathematical apparatus to provide many practical methods with such guarantees together. This is why many performant works do not have them. Examples include: AlexNet [1] from computer vision, Transformer [2] from language modeling, DDOM from MBO [4], PPO [7] from reinforcement learning, StyleGAN-XL [8] from generative modeling, etc. The first of the cited by the reviewer papers, BootGen, does not provide such guarantees either.
>
>
> ## Questions
>
> 1. How to choose the FGM configuration for a new problem?
>
> > We recommend the procedure we gave in the answer to the first critique. Additionally, we believe that, in practical applications, a method similar to how MBO research is conducted would be helpful in selecting the hyper-parameters. That is, for a given dataset, a surrogate oracle $\hat{y}(x)$ can be trained on the real-world labels $y(x)$. Then, the labels $y$ can be replaced with predictions $\hat{y}$ made by the surrogate. One can then train an MBO algorithm with respect to these new labels and see which hyper-parameters perform best.
>
> ## References
>
> [1] A. Krizhevsky et al. 2012; ImageNet Classification with Deep Convolutional Neural Networks
>
> [2] A. Vaswani et al. 2017; Attention Is All You Need
>
> [3] J. Kaplan et al. 2020; Scaling Laws For Neural Language Models
>
> [4] S. Krishnamoorthy et al. 2023; Diffusion Models for Black-Box Optimization
>
> [5] J. Ho et al. 2020; Denoising Diffusion Probabilistic Models
>
> [6] B. Trabucco et al. 2021; Conservative Objective Models for Effective Offline Model-Based Optimization
>
> [7] J. Schulman et al. 2017; Proximal Policy Optimization Algorithms
>
> [8] A. Sauer et al. 2022; StyleGAN-XL: Scaling StyleGAN to Large Diverse Datasets

---

> > ### Comment · Reviewer_JStj · 2024-11-25
> >
> > Thank you for your response, and apologies for the late reply.
> >
> > > In our experiments, we found the following scheme to work generally well
> >
> > Thanks for the details, These would be useful to include in the paper.
> >
> > > In fact, this is quite normal for works that introduce new models ...
> >
> > With all due respect, I do not think the absence of ablations on other ("seminal") papers justifies the lack of the ablations in this paper. I believe the question of hyperparameters is critical here - the hyperparameters represent the underlying structure to assume in the data, which to me seems more important than the dropout rate in AlexNet. While I understand that running the ablations comes at increased cost, I think it is reasonable to have these ablations in the paper.
> >
> > > We have added results of the more recent SOTA diffusion-based method DDOM
> >
> > Thanks for adding the baseline. I am a bit surprised by the results - which show that COMs outperform DDOM contrary to the results in the DDOM paper. Do you have any intuition why this is happening?
> >
> > Additionally, while going over the new results I noticed some discrepancies in the experiments which I missed earlier. Mainly, the authors report the mean top-10 score, whereas the norm in MBO is to report the 100th percentile (max), including in the Trabucco et al. 2021 which the authors cite. Moreover, this paper uses 1000 samples whereas Trabucco et al. use 128. Could you please elaborate on these choices? Why deviate from the standard setup?
> >
> > > We understandand if the reviewer would like to see computational complexity analysis, but we do not think that lack of it is a unique flaw of our paper
> >
> > With all due respect, I do not believe this is an unreasonable request. For a reader, it provides important context, which can influence the choice of method. A simple table with the runtime of each method would suffice and provide useful information.
> >
> > > However, modern deep learning - which we rely on - lacks mathematical apparatus to provide many practical methods with such guarantees together
> >
> > I understand that providing theoretical guarantees is challenging in deep learning. However, the point of the review is to highlight the weaknesses and in my view this is a valid one (which also applies to other work). Also I find it rather dismissive of the authors to claim "modern deep learning - which we rely on - lacks mathematical apparatus to provide many practical methods with such guarantees together" when there has been much progress in the area of DL theory.

---

> > > ### Author Response · Authors · 2024-11-26
> > > **Answering the remaining questions**
> > >
> > > **Critique about insufficient evaluation of hyper-parameters**
> > > > We agree with the reviewer that ablations about the imposed structure (clique decomposition) should be included in the paper. This is why we have section *5.2 Ablations*. Beyond that, we do not know what the reviewer means by ablations about the underlying structure. Moreover, we are afraid that, whatever we would be able to find with a handful of experiments would not be of great use, since the optimality of hyper-parameters is problem-dependent. We would not be able to find *Scaling Laws for Cliqueformer* in this paper, and unless a specific task with enormous dataset size is available (like in language modeling), we will not be able to find general results about these hyper-parameters soon. Pardon us, but we feel as troubled as if the reviewer asked us for the optimal hyper-parameter configuration of an MLP or a CNN.
> > >
> > > **Why are you Reporting top-1% instead of top-1 of 128?**
> > > > The reason is simple: to make the reported results more robust to variance. Note that COMs reports top-1 of 128 scores, while we report the mean of top-10 out of 1000. Thus, we end up reporting *approximately* the same statistic, which is the estimate of the mean of the top-1% bin of the score distribution, and the way to report it more robustly is to take more samples, and average the top-1% of them. Even more simply and practically, reporting top-1 (or generally *max*) is susceptible to lucky examples, and provides an easy way to hide that an algorithm produces many poor designs. Perhaps it is a good moment for us to point to the *validity* scores in Table 1 in blue - while COMs is known to achieve good top-1 scores, it can do a poor job at even generating valid designs. Lastly, it seems a bit too strong to say that top-1/128 is the standard as wet-lab engineers often prefer to study more than one model output (for example, [1] evaluate top 10% of model prediction recalls). Also, CbAS reports various percentiles of scores of $100$ designs (not 128), 100th being only one of them.
> > >
> > > **Why does COMs outperform DDOM? In DDOM paper it was the other way around?**
> > >
> > > > Thank you for making this observation and asking this question. We believe that in strengthens the point that we made above: the more samples we take and average over, the better. Note that both COMs and DDOM follow the same evaluation scheme (top 1/128) and report mutually "contradictory" claims - in Trabucco et al. 2021, COMs outperforms Gradient Ascent, while in DDOM, it is Gradient Ascent that outperforms COMs in terms of average rank. In our experiments the difference between COMs and Gradient Ascent is clearer and our results tilt towards COMs. Note that it tends to work better, thought not *well*, than DDOM mainly on synthetic tasks though, possibly due to its conservative regularizer. These tasks were not used in the DDOM paper, and hence the average results in this paper are different than in DDOM. Because of this, and because of the randomness of the experiments, we see no contradiction.
> > >
> > > **There has been enough progress in deep learning theory for Cliqeuformer to come with theoretical guarantees**
> > >
> > > > We tried to find papers on convergence of transformers. However, we did not find papers that would apply to models that we ended up using. For example, [2] studies the convergence of 1-block (we have several blocks) encoder-only transformer (we have both an encoder and a decoder) in predicting a scalar label with MSE (we use transformers for auto-encoding). Even more recent [3] trains a single-layer transformer as well, with a target of predicting a value of a linear function (our targets are highly non-linear). They also assume a specific sparsity form of their weight matrices (see Equation 2). Even in these simplified settings, it takes an entire theory paper to deliver the desired results. While being clearly great, these works do not provide us with tools that would enable us to prove convergence of Cliqueformer in the remaining half-a-page we have. That would probably be another paper, and we are afraid that this feat will take theoreticians a bit more time. Is there anything, either on the theoretical side of our method that is in our current reach, or in related work, that the Reviewer can see but we can't? Does the Reviewer have a proof sketch in mind? If not, we would like to state the obvious which is that we conduct, mainly, empirical research, and we believe that there is place for such work at ICLR.
> > >
> > >
> > > **References**
> > >
> > > [1] P. Notin et al. 2023; ProteinGym: Large-Scale Benchmarks for Protein Design and Fitness Prediction
> > >
> > > [2] Y. Wu et al. 2023; On the Convergence of Encoder-only Shallow Transformers
> > >
> > > [3] Y. Huang et al. 2024; In-context Convergence of Transformers

---

> > > > ### Author Response · Authors · 2024-11-26
> > > >
> > > > Dear Reviewer,
> > > >
> > > > Folllowing your suggestion, we added computational complexities of methods used in experiments (and of Cliqueformer's) in terms of the number of FLOPs, for both the training step and the design optimization phase, in Table 2, Appendix C. We hope that this addresses the Reivewer's concerns.
> > > >
> > > > In addition to the strong results achieved by our model and insights we obtained while obtaining them, in our rebuttal, we added new baseline evaluations, more related work, extra empirical investigation of learned representations, the computational complexity table, and answered your questions in our comments, with support coming from literature. As of now, the Reviewer keeps their score at level 3. We would love to hear if there is something we can do to lift this score or there is something fundamentally flawed in our paper?
> > > >
> > > > We send the Reviewer the kindest regards,
> > > >
> > > > Authors

---

> > > > > ### Comment · Reviewer_JStj · 2024-12-02
> > > > >
> > > > > Thanks for your response, and sorry for my late reply.
> > > > >
> > > > > > This is why we have section 5.2 Ablations. Beyond that, we do not know what the reviewer means by ablations about the underlying structure...
> > > > >
> > > > > I think Section 5.2 is a great start, but it studies only the effect of the number of cliques. I think a similar analysis for d_clique, and the number of design steps would also be useful.
> > > > >
> > > > > > since the optimality of hyper-parameters is problem-dependent
> > > > >
> > > > > My motivation to suggest more careful ablations is not to find the optimal hyperparameters, but rather to understand the effect of the choice of parameters on the performance.
> > > > >
> > > > > > Lastly, it seems a bit too strong to say that top-1/128 is the standard as wet-lab engineers..
> > > > >
> > > > > I agree that the top-1 metric is not ideal but I believe it is important to also report it in addition to the mean of the top 10% since much of the prior work in the field uses the metric. Especially considering that the experiments in the paper only consider a couple of MBO baselines.
> > > > >
> > > > > > There has been enough progress in deep learning theory for Cliqeuformer to come with theoretical guarantees
> > > > >
> > > > > Perhaps there is some miscommunication - this was not what I was suggesting. My comment was that the method does not come with theoretical guarantees (which can be important in applications where MBO approaches are used) and this limitation should be acknowledged in the paper. To be very clear, I think it is completely okay for an approach to not have such guarantees but I do think it is a limitation which should be discussed in the paper.
> > > > >
> > > > > > Folllowing your suggestion, we added computational complexities of methods used in experiments (and of Cliqueformer's) in terms of the number of FLOPs, for both the training step and the design optimization phase, in Table 2, Appendix C. We hope that this addresses the Reivewer's concerns.
> > > > >
> > > > > Thanks for adding these details. Sorry, this was not clearly specified in my initial review but mentioned in my comment - more than the computational complexity - what I think is more useful for people using your method is the actual wall-clock runtime.
> > > > >
> > > > > I apologize to the authors for responding so late. I do believe there are interesting ideas presented in the paper but I encourage the authors to take the feedback into account since the paper could certainly be improved!

---

> > > > > > ### Author Response · Authors · 2024-12-03
> > > > > > **Discussion quality**
> > > > > >
> > > > > > Dear Reviewer,
> > > > > >
> > > > > > Thank you for this comment, as well as for clarification of your concerns. We can include the wall-clock time table in the final version of our paper if the reviewer believes it would improve the paper's quality. We can also add a table with results with top-1/128 evaluations - this is not really a problem, but we simply wanted to enhance our evaluations with additional robustness, so that we avoid "surprises" like that one we discussed before, involving COMs and DDOM papers. Also, we would like to point out the typo/misunderstanding made by the reviewer above - we evaluate top 1%, not top 10%.
> > > > > >
> > > > > > However, as the paper revision deadline is past due, we cannot edit the paper now. Thus, the only thing we can do is to respond to the reviewer, whom we encourage to evaluate the quality of the work he/she put into grading us and providing us with constructive and fair feedback. As the reviewer noted themselves, their comments were not concrete enough, and led us not to do what the reviewer requested (like in the case of run time). We are greatly confused by the reviewer's concern about theoretical guarantees also. Nothing indicated that the reviewer accepts that lack of theoretical guarantees is OK if mentioned - the reviewer's comment about progress in DL theory suggested otherwise. Since we took this discussion seriously, we invested time into additional research, and the citations above are the outcome of this. As we mentioned, they don't bring us anywhere close to equipping our paper with guarantees. As for the current narrative of the reviewer (about listing lack of theoretical guarantees as limitation), we read the (few) papers that the reviewer cited. **None** of them comes with a single theorem, and **none** of them lists lack of theoretical guarantees as a weakness. In combination with the reviewer's unclear stance on this throughout the discussion period, we consider this comment as adversarial.
> > > > > >
> > > > > > Thank you for making it clear that $d_{clique}$ is what the reviewer would like to see to be ablated for. We can add a sweep through its values in the final version, but as of now, there is nothing we can do with the current submission. Again, the reviewer did not make it clear during the 2 weeks when we could add technical work to our submission that this ablation is what the reviewer thinks is important. Even now, *why does the reviewer think it's important?* Typical for this discussion, there is little commitment from the reviewer's side to comments made about our work.
> > > > > >
> > > > > > We also need to point out - for the reviewer and everyone that reads this - that the runtime analysis, whose lack the reviewer listed among the biggest weaknesses of our paper, is actually unimportant in the context of offline MBO. That's because its goal is to skip weeks or months of wet-lab experiments, and thus the difference of a few minutes or hours that we spend training a model are negligible. This is why COMs, that is notoriously "slow" to train due to its inner adversarial loop, does not compare run times, nor does **any** of the papers that the reviewer cited. Thus, again, we consider this comment arbitrary at best.
> > > > > >
> > > > > > The reviewer also pointed out 4 papers we should have compared against. 2 of them are developed for biological sequence design - a problem narrower than the setting we tackle. The other 2 constitute a valid critique. Nevertheless, they are very new papers, boasting **jointly** 18 citations (as of Dec 02/2024), which is **less than half** of DDOM (39) which we compare to, and outperform, which the cited papers omit in their experiments. Hence, while we don't claim that our baseline comparison is extensive, we do not believe that we violated common evaluation standards.
> > > > > >
> > > > > > The discussion omitted contributions of our papers, such as: (1) theorem about non-uniqueness of FGM; (2) clear presentation of deduction that led to Cliqueformer, which (3) offers an approach to solve MBO solely through use of DL tools, such as decomposition of the predictive network, or the VIB loss - without complex algorithmic techniques, such as COMs inner loop or ICT co-teaching; (4) the surprising role played by AdamW optimizer; and (5) the strong empirical performance. Meanwhile, with the above summary of the reviewer's comments, which contained baseline mis-recommendation, lacked evidence in form of papers or strong argument, were vague and infrequent, we believe that the reviewer, who gave us a score of 3, did not stand to the standard he/she expected us to.
> > > > > >
> > > > > > Thank you for participation in the review of our paper,
> > > > > >
> > > > > > Authors

---

> > > > > > > ### Comment · Reviewer_JStj · 2024-12-03
> > > > > > >
> > > > > > > I understand that the paper cannot be updated but I trust that the authors will add the wall clock runtime and ablations on d_clique (and number of design steps which they might have missed in my comment) to the next revision - I have raised my score conditioned on these changes. I appreciate the authors constructive criticism but disagree with their characterization of the discussion.
> > > > > > >
> > > > > > > > the reviewer's comment about progress in DL theory suggested otherwise
> > > > > > >
> > > > > > > The comment was just in response to the author's statement in the initial rebuttal which was dismissive of the entire area of DL theory as indicated in my original comment.
> > > > > > >
> > > > > > > > In combination with the reviewer's unclear stance on this throughout the discussion period, we consider this comment as adversarial.
> > > > > > >
> > > > > > > With all due respect, my initial review clearly mentioned the issue was the authors not discussing related work sufficiently in their rebuttal and not comparing to it. I did not claim that the papers are perfect. My suggestion was (and remains) that the authors should be upfront about the lack of theoretical guarantees (and this applies to the related work too - but I do not have the ability to change those papers).
> > > > > > >
> > > > > > > > why does the reviewer think it's important?
> > > > > > >
> > > > > > > The method builds upon underlying FGM structure of the function being optimized. In practice, this structure is typically unknown. For anyone applying this method to a new problem having an ablation showing the effect each of the hyperparameters has on the performance can be very helpful. Moreover, it tells us something more about how/why the proposed approach works.
> > > > > > >
> > > > > > > > That's because its goal is to skip weeks or months of wet-lab experiments, and thus the difference of a few minutes or hours that we spend training a model are negligible.
> > > > > > >
> > > > > > > I do tend to agree with this sentiment but I believe this maybe useful information for someone trying to use this method on a new problem to get a rough approximation for how long the experiments would take.
> > > > > > >
> > > > > > > > This is why COMs, that is notoriously "slow" to train due to its inner adversarial loop, does not compare run times, nor does any of the papers that the reviewer cited. Thus, again, we consider this comment arbitrary at best.
> > > > > > >
> > > > > > > Again, with all due respect, I do not think citing other papers not performing a proper analysis is convincing (neither did I mention the cited papers for their attention to detail).
> > > > > > >
> > > > > > > > The reviewer also pointed out 4 papers we should have compared against
> > > > > > >
> > > > > > > My original review cited the papers as examples of prior works which were not discussed in the paper.
> > > > > > >
> > > > > > >
> > > > > > > > Nevertheless, they are very new papers, boasting jointly 18 citations (as of Dec 02/2024), which is less than half of DDOM (39) which we compare to, and outperform, which the cited papers omit in their experiments.
> > > > > > >
> > > > > > > With all due respect, I do not believe number of citations should dictate what baselines are considered for a method (metrics like citations are influenced by a lot of factors other than the content of the paper). Secondly, the papers I cited appeared on arXiv earlier than or around the same time as DDOM (enough to be considered concurrent work), and all of them appeared on arXiv and were published almost one year ago - so I do not agree that they are very new papers.
> > > > > > >
> > > > > > > > The discussion omitted contributions of our papers
> > > > > > >
> > > > > > > My initial review specifically mentioned the VIB-inspired loss and FGM connections in the strengths. And a lot of my questions were to understand the empirical performance. So I am disappointed that the authors claim these points were not discussed.

---

### Official Review · Reviewer_eafL · 2024-11-03

**Soundness:** 3
**Presentation:** 3
**Contribution:** 2
**Rating:** 5
**Confidence:** 3

**Summary:**

This paper aims at proposing a model capable of learning the structure of a model-based optimization task under a functional graphical model's framework. The evaluations include a black box optimization task in the 10-s of dimensions range, and some chemical and genetic design tasks. Unlike previous work, the FGM discovery is merged into the model's learning, and the marginal distributions of the cliques were enforced to mimic a standard normal by minimizing a KL divergence.

**Strengths:**

1. The notations and model definitions were well-described. I had an easy time understanding the approach, and the explanations were easy on the mind.

2. The intuitions in the paper are stated clearly, and the desideratums were easy to follow.

3. The figures seemed effective at explaining the related concepts. Figure 4 presents a clear summary of the model, Figure 3 clearly demonstrates the problem with FGM choices, and Figures 1 and 2 are also helpful.

4. A limited set of ablation studies show the effect of the number of cliques on the downstream score.

**Weaknesses:**

1. As is, I fear the introduction and model description sections are quite disconnected from the experiments. The authors spend a substantial amount of time in Sections 1-4 building intution on how the proposed model should work, what features are desired, and so on. The experiments don't do much more than presenting some benchmarks. This makes it difficult for me to verify that the gained improvements are actually the result of the claims and contributions, rathen than being side effects of auxillary design choices. For instance,

  * Was the transformer architecture adaptation the main source of improvement?

  * Was the VIB loss effective at all? What happens if it is removed? How much of the improvements could be attributed to this structuring of the latent space? Is there data demonstrating the inherent risk by enforcing this loss too weakly or too forcefully?

  * Were the FGM components of the model effective at "exploiting the structure of the target black-box function"? Were there any practical results, *in this paper*, answering this specific claim?

  * Did the method actually result in better coverage of the cliques data, and contributed to the distribution-shift issue? Figure 1 alone seems rather insufficient to address this broad question.

2. The paper is based more around "proposing a solution", than the proper scientific steps of (1) identifying a narrow, previously unknown, problem in the prior methods or datasets, (2) showing evidence about the existence of the problem, and then (3) proposing a solution and demonstrating its efficacy at addressing this issue.

3. The experimental breadth and the considered methods are awfully inadequate. At this point, there are tens, if not hundreds, of other successful MBO methods, and reward weighted regression and grad asc, for instance, are irrelevant and not remotely representatives of the state of the literature. Since there is a strong emphasis on (1) RL strategies for MBO, and (2) the fact that many of the baselines are sequence-based, the authors should add relevant baselines such as the following:

  * DynaPPO (Model-based reinforcement learning for biological sequence design, ICLR 2020)

  * Ada-Lead (https://arxiv.org/pdf/2010.02141) in FLEXS (https://flexs.readthedocs.io/en/latest/index.html) has a standard implementation and is another contender.

  * PEX (Proximal Exploration for Model-guided Protein Sequence Design, ICML 2022)

  * FBGAN (Feedback gan (fbgan) for dna: A novel feedback-loop architecture for optimizing protein functions)

  * Other rather weak and old baselines are Cbas, Dbas, CEM-PI, etc.

4. Theorem 1 is a restatement of a basic result in batch reinforcement learning, and offers a rather weak upper bound. While such results are helpful in deriving RL theory, they're nowhere close to being practical; the $C_{stat}$ and $C_{cpx}$ can be arbitrarily large, and the maximum term is almost impossible to estimate. Since this bound is practically almost never tight, it makes the claims and contributions in this paper less verifiable.

**Recommendation**

I feel ambivalent about this paper. On one hand, I see that the authors clearly spent reasonable time and efforts developing an idea, implementing the model, and writing the paper. There is clearly some quality effort spent here. On the other hand, the scientific skeleton of the paper and the supporting evidence is significantly lacking, as I described earlier, and just giving a pass here seems like quite the negligence on my part.

After much delibration, I kindly decided to give the work a weak rejection for now. That being said, I'm keeping an open mind, and I would be happy to raise my score if the authors deliver a strong rebuttal, i.e., a rebuttal that adds significant amounts of supporting experiments and data to the paper and fills up the glaring void. I hope the authors understand that they need much more experimental results in this paper for it to be published at a top-tier venue.

**Questions:**

Please see the points raised in the weaknesses section.

## Minor Comments

1. Line 249: "distrituion" should be "distribution".

2. Line 731: "failute" on should be "failure".

3. Line 80: "More results in Section 5" is not a complete sentence.

4. Line 745: "We use dropout of rate 0.5" should be replaced with "We use a dropout rate of 0.5"​.

5. The abstract does not seem fitting. The content prior to the "In this paper" sentence, i.e., Lines 11-25, do not strike me as abstract material, and are best suited elsewhere in the paper. For instance, I don't consider explaining "why RL strategies generally work better on MBO problems" a priority for the abstract. Only Lines 26-33 seem specifically relevant to the paper's contributions.

---

> ### Author Response · Authors · 2024-11-19
>
> ### Critique
>
> 1. The paper does not follow the *proper* scientific steps of (1) identifying a narrow problem in prior art, (2) proving its severity, and (3) proposing a solution to it. Is is, instead, focused on proposing a solution.
>
> > The approach the reviewer delineated sounds perfectly reasonable and we are sorry that the reviewer felt forced to give such a comment. However, we must contest it. The path that the reviewer offered is not the only valid way to conduct research. In fact, much of improvement in deep learning comes from projects that started by approaching a considered task from a new angle. Examples include AlexNet [1], Transformer [2], Diffusion Models [3], etc. Nevertheless, our project did tackle a specific problem: most of the research in MBO focused on algorithmic tenchiques, employing simple network architectures, to solve design tasks. Our goal was to show that such intricate algorithms are not necessary with careful adoption of the most performant deep learning architectures. And indeed, with Cliqueformer, we were able to optimize our designs with vanilla AdamW optimizer.
>
> 2. The experiments focus on the final performance of the model. Specific components of it lack empirical justification.
>
> > To begin with, we agree that we did not include experimental evidence for every compoenent of our architecture. However, in Section 3, we did propose reasoning that invokes evidence from prior work and our theoretical results. Note that Desideratum 1 is supported by (prior) Theorem 1 and our Theorem 2. Desideratum 2 also follows from Theorem 1, as well as it is intuitively clear: optimizing designs with respect to a model that was trained on too narrow of a set of designs is futile; the only question is if we can be less aggressive in enforcing the coverage requirements. Theorem 1 answers this question affirmatively, and ablations in Figure 5 provide empirical evidence for it. Additionally, in the revised version we add a Transformer baseline (without the regularized latent space and the clique decomposition) to our experiments and obtain largely inferior performance. We believe it provides additional support for our architecture.
>
> > We understand the reviewer's uncertainty about the empirical evidence for all components of Cliqueformer. Similarly to the previous question, however, we want to highlight that introducing a new approach to a considered task makes authors spend more time describing the approach - the remaining questions show we could spend even more time on that - and thus less time to properly study individual pieces. We want to be very honest: studying each step in isolation would be impossible here - see for example Line 47 in *models/cliqueformer.py*, where we state the best order of layers used in the implemented step. Studying each such piece properly would easily make us run out of space. Note that, for example, the Transformer paper [2] does not study much how important the order of layers is in a single block, or the attention head dimension is ([4], Fig 5, later showed that it is not that important). Also, the Diffusion Models paper [3] introduces many components, like the variance schedule, that it gets to work without a deeper analysis - this arrived later in follow-up works [5]. Similarly, we believe that a deeper empirical and theoretical analysis of Cliqueformer-like methods is an exciting avenue of future work.
>
> 3. Did the method actually result in better coverage of the clique data?
>
> > If we understood the question correctly, it actually asks about the effectiveness of VIBs/VAEs. To provide an instance-level piece of evidence, we ran an empirical study of the distribution of design representations in the revised version (Appendix B). The results indicate that the latent space did attain good coverage.
>
> 4. The current experimental breadth is awfully inadequate and the authors should run more baselines, e.g. DynaPPO, PEX, etc.
>
> > We are sorry that the reviewer feels so strongly about our experiments. Nevertheless, they are right that we shall update our range of baselines. In the revised version, we benchmarked a recent diffusion-based DDOM [6], and gradient ascent with the transformer backbone. When it comes to the methods suggested by the reviewer, they make us wonder if the problem setting of our paper is clear. Our work focuses on **offline** model-based optimization. Instead, the proposed methods come from the literature on protein/DNA-sequence design specifically. Crucially, they don't assume the offline scenario either - but the opposite online one. For example, in DynaPPO, a reinforcement learning-like ability to recollect new data is assumed (see Line 7 in Algorithm 1 of the paper). Also, PEX literally stands for proximal **exploration**, and contains wet-lab experiments as sub-routine (Algorithm 1). Could the reviewer re-consider our grade from the perspective of general-purpose offline MBO?

---

> ### Author Response · Authors · 2024-11-19
>
> ### Questions
>
> 1. Was the transformer backbone the main source of improvement?
>
> > Thank you for pointing this out. To answer this question unequivocally, in the revised version, we evaluated gradient ascent with the transformer backbone. It achieves only slightly better performance than gradient ascent with MLP, and thus the answer to the reviewer's question is *no*. Similarly, as we found in Section 5.2, appropriate setting of the clique decomposition has big impact, even though the transformer backbone is employed for all runs. However, as we explained in lines 257-260 of the original submission, as well as we responded to Reviewer XTE4, the expressive transformer backbone is essential to enable learning representations that satisfy our desiderata.
>
> 2. Have the authors provided evidence that the proposed FGM compoenents were effective at *exploiting the structure of the target black-box function*?
>
> > Yes, we have. In addition to strong performance of the method as a whole, we swept over different configurations of the decomposition in Section 5.2. In each of the three sweeps, we covered the case $N_{clique}=1$, which corresponds to lack of the decomposition. In each of the tasks, this configuration was sub-optimal, loosing to configurations with many cliques.
>
>
> ### References
>
> [1] A. Krizhevsky et al. 2012; ImageNet Classification with Deep Convolutional Neural Networks
>
> [2] A. Vaswani et al. 2017; Attention Is All You Need
>
> [3] J. Ho et al. 2020; Denoising Diffusion Probabilistic Models
>
> [4] Y. Song et al. 2021; Score-Based Generative Modeling Through Stochastic Differential Equations
>
> [5] J. Kaplan et al. 2020; Scaling Laws For Neural Language Models
>
> [6] S. Krishnamoorthy et al. 2023; Diffusion Models for Black-Box Optimization

---

> ### Author Response · Authors · 2024-11-26
> **Reminder**
>
> Dear Reviewer,
>
> In addition to the rebuttal we responded to you above with, we added a new section *More Related Work* in Appendix D, in which we described the position of our paper with respect to works you have cited, such as DynaPPO and PEX. You have hinted a possibility of raising your score, given a strong rebuttal. As of now, we have run additional baseline experiments, including an ablative Transformer baseline, conducted extra experimental investigation in Appendix, upgraded our figures, responded to your questions, and added related work in the appendix. We would, therefore, greatly appreciate your updated feedback.
>
> Best wishes,
>
> Authors

---

> > ### Comment · Reviewer_eafL · 2024-11-27
> > **Response to the Authors**
> >
> > I would like to thank the authors for their rebuttals and efforts, and extend an apology for the delayed response.
> >
> > I've read the other reviewers' comments, the authors' rebuttals, and the revisions made in the paper. The results about the new diffusion method DDOM is interesting. However, it is not sufficient; my main concern about the experimental breadth and depth of the paper remains largely unaddressed. I'm afraid I share Reviewer JStj's stance as *I also find the authors' insistence on justifying the lack of ablations by the absence of ablations in other papers odd*.
> >
> > Furthermore, I don't find the reasoning for dismissal of other baselines valid. Almost all MBO papers rely on an offline evaluation protocol using a pre-defined dataset. This is a common practice due to physical costs. All such frameworks can be modified to accommodate an offline setup and therefore be amenable to evaluation in this paper. In fact, this practice is quite typical in optimization areas other than MBO, such as multi- and contextual-armed bandits in reinforcement learning.
> >
> > If the authors still felt the proposed methods or comments were inapplicable from the reviewers, I would have still been happier if they suggested more methods, datasets, and studies and evaluated them more carefully. Offline zero-order optimization has been studied for decades with many flavors and there are more strong baselines and datasets than I could ever count. The addition of DDOM and the transformer baseline certainly help, but **the experimental results are still significantly limited** and I find the additions insufficient.
> >
> > Unfortunately, as is, the paper mainly relies on a single table of experimental results and a small number of baselines. The extreme statistics reported in the paper were also under question, and I'm not convinced by the authors' rebuttals. The number of negative infinity results may seem as a strength to the authors, but it makes the reader question whether the methods were truly evaluated in their best light and/or the evaluation protocol was chosen properly. The lack of supporting evidence and ablations makes these issues much worse, and the authors' excuse for space limitations is not convincing.
> >
> > All in all, I truly believe the paper could still use another round of review and more time and efforts to solidify its empirical evaluations' standing; this work could certainly become a much stronger submission for a close-by ML conference venue.

---

### Official Review · Reviewer_XTE4 · 2024-11-04

**Soundness:** 3
**Presentation:** 3
**Contribution:** 3
**Rating:** 6
**Confidence:** 4

**Summary:**

The paper proposes Cliqueformer, a new method for offline model-based optimization. The method is built on the idea of functional graph model (FGM) which decomposes the input space into cliques that contribute independently to the function. The paper proposes to learn a latent representation of the input space, imposes FGM cliques on this space, and simultaneously trains regression models on the latent cliques. To perform optimization, the model takes gradient steps of the regression model against some initial latent representations and decodes back to the original input space.

**Strengths:**

The paper presents a novel method for offline model-based optimization. The idea of learning a latent space and imposing FGM on it is interesting. I also like the simplicity of the model and the clear writing of the paper. The empirical performance of the proposed method is strong against different baselines.

**Weaknesses:**

- Looking at Figure 5, the performance varies significantly with different values of cliques, and there is not a universal value across different datasets. How did the authors pick the value in an offline setting where you're not allowed to query the oracle?
- The baselines seem quite outdated. Have the authors tried comparing with more recent baselines such as generative methods using transformers or diffusion models [1, 2, 3, 4]? Some of these papers should also be cited and discussed in the paper.
- The Superconductor task uses a surrogate function and not the true oracle, which was found to not be reliable in [4].
- Minor comment: Algorithm 1 should remove the training process of Cliqueformer for better clarity.

[1] Krishnamoorthy, Siddarth, Satvik Mehul Mashkaria, and Aditya Grover. "Generative pretraining for black-box optimization." arXiv preprint arXiv:2206.10786 (2022).

[2] Krishnamoorthy, Siddarth, Satvik Mehul Mashkaria, and Aditya Grover. "Diffusion models for black-box optimization." International Conference on Machine Learning. PMLR, 2023.

[3] Chen, Can, et al. "Bidirectional learning for offline infinite-width model-based optimization." Advances in Neural Information Processing Systems 35 (2022): 29454-29467.

[4] Nguyen, Tung, Sudhanshu Agrawal, and Aditya Grover. "Expt: Synthetic pretraining for few-shot experimental design." Advances in Neural Information Processing Systems 36 (2023): 45856-45869.

**Questions:**

- Why does Cliqueformer suffer less from distribution shift problems? Can adding an additional conservative regularization benefit the model?
- Why does the decoder have to be a transformer model? What if we use a linear or shallow MLP prediction head instead? This is equivalent to using the last layer of the transformer backbone for the latent representation which is the common practice in many domains.

---

> ### Author Response · Authors · 2024-11-19
>
> ### Critique
>
> 1. The baselines seem quite outdated. Shouldn't the authors compare against more recent methods, like the diffusion-based DDOM?
>
> > The author is right in pointing this out. We implemented and evaluated DDOM on our set of tasks and added the results to the revised paper. We also evaluated gradient ascent with Transformer backbone to provide evidence of the impact of the novel mechanisms in Cliqueformer.
>
> 2. Superconductor task uses a surrogate function instead of the true oracle, which was found not to be reliable.
>
> > It is true that a surrogate oracle is used in Superconductor, as well as in all continuous tasks. This is an absolute necessity, since evaluation of continuous designs would require synthesizing them in a wet lab. The accuracy of this oracle does not really matter that much since the training data are also labeled with it, which keeps the target function of our MBO algorithm consistent. Lastly, we investigated the evidence of that inaccuracy for Superconductor provided by the cited work of Nguyen et al. 2023 and found out that only a line plot, without alpha transparency, of several of thousands of examples was shown. This makes the graph unreadable and potentially misleading. On the contrary, it is known that this oracle achieves a very high Spearman-rank correlation of 0.927.
>
> ### Questions
>
> 1. Figure 5 shows that the performance varies with different values of $N_{clique}$. How did the authors pick this value?
>
> > Finding a hyper-parameter setting scheme was part of our work. We conducted this investigation on LRBF tasks which, being synthetic, do not require that we collect external data, but instead allow us to study problems of arbitrary dimensionality. We have found the following scheme to work well reasonably well: we fix $d_{knot}=1$ and, *(1)* for tasks where $dim(x)<100$, we fix $d_{clique}=3$ and set $N_{clique}$ so that $dim(z)=N_{clique} \cdot (d_{clique} - d_{knot}) + d_{knot} \approx dim(x)$; *(2)* for tasks where $dim(x) \geq 100$, we fix $d_{clique}=6$, and repeat the procedure to find $N_{clique}$ - the reason for doing that is to decrease the cost of computing attention layers which is quadratic in $N_{clique}$. In Table 1, we made a single exception from this rule - for Superconductor, where another setting was indeed much better.
>
> > Generally, however, we believe that these values can be tuned in practice, in a very similar fashion MBO researchers evaluate their methods in papers. Namely, one can pre-train a surrogate oracle on a *large* dataset, label the data with the surrogate, and train the MBO model on a subset of the data with the new labels. This way, one can sweep over hyper-parameters to tackle the real dataset with.
>
> 2. Why does the decoder have to be a transformer? What happens if it is replaced with an MLP?
>
> > In our early experiments, we tried to implement our model with MLPs. This proved very difficult. We believe that the reason for that is the difficulty posed onto the model - to learn representations that provide good reconstructions, satisfy Gaussianity properties, and can predict the target function. After we switched to Transformers, we could learn such representations. The only place where we did not see benefit of such a change was the predictive model $f_{\theta}(z)$, where we stack with MLPs.
>
> 3. Why does Cliqueformer suffer less from distribution shift problems? Can adding an additional conservative regularization benefit the model?
>
> > Cliqueformer has an auto-encoder, as a sub-module, which produces representations $z(x)$. The representations are regularized with a KL divergence loss, so that a distribution of such representations can correspond to a single design $x$ [1]. The reviewer can think of it as partitioning the $z$-space into pieces which correspond to designs $x$. Note that we optimize our designs by optimizing representations $z$ that are later decoder to actual designs $x$. A helpful property is that, while a single gradient step cen create an invalid design, it is difficult to move from a valid-design region of $z$-space to an invalid region. In order to stay extra catious, though, we use weight decay while optimizing $z$: that is, we bring $z$ closer to $0$ with every gradient step. The reason for that is that, under standard Gaussian priors, the region of $z$-space near $0$ corresponds to valid designs, thus preventing the distribution shift. We hope that this helped. If not, would the reviewer be so kind to ask a follow-up question?
>
> [1] D. Kingma and M. Welling 2013; Auto-encoding Variational Bayes

---

> > ### Comment · Reviewer_XTE4 · 2024-11-23
> >
> > Thank you for your rebuttal. I'd like to keep my original score and lean towards accepting the paper.

---

### Official Review · Reviewer_98z9 · 2024-11-04

**Soundness:** 2
**Presentation:** 3
**Contribution:** 2
**Rating:** 6
**Confidence:** 3

**Summary:**

This work proposes a new architecture, Cliqueformer, for solving model-based optimization (MBO) using the functional graphical model (FGM) framework. Cliqueformer subsumes the graph discovery step within its architecture, rather than using the graph discovery heuristic from the previous work (Grudzien et al., 2024). The VIB-style objective further ensures broad coverage of each clique’s distribution, enhancing stability in optimization. Experimental results demonstrate that Cliqueformer outperforms previous methods across a range of tasks.

**Strengths:**

- Cliqueformer effectively addresses the limitations of FGM by eliminating the need for explicit graph discovery steps, while still maintaining the scalability advantages of MBO with FGM.
- The proposed method is carefully designed based on theoretical observations
- It demonstrates significant performance improvements over COMs across a wide variety of tasks.
- well-written with clear motivation

**Weaknesses:**

While the paper makes significant contributions, some claims might be somewhat overstated:

- The authors state that "we consistently obtained good performance by setting the clique size to $ d_{\text{clique}} = 3 $." However, the experimental results suggest that the choice of $ d_{\text{clique}} $ or $ N_{\text{clique}} $ significantly affects performance. For instance, Figure 5(b) shows that only $ N=4 $ achieves better performance than $ N=1 $ (assuming $ N=1 $ represents the whole graph), and depending on how $ N_{\text{clique}} $ is set, Cliqueformer sometimes fails to achieve scores higher than 1.
- The claim that "Cliqueformer achieves state-of-the-art performance" may be premature. Since some recent studies are not included as baselines (e.g., BDI [1], BootGen [2], or the baselines in Design-bench [3] for TFBind and superconductor), and because the scores are normalized differently, it's challenging to assert that Cliqueformer achieves state-of-the-art results conclusively.

[1] Chen, Can, et al. "Bidirectional learning for offline infinite-width model-based optimization." Advances in Neural Information Processing Systems 35 (2022): 29454-29467.

[2] Kim, Minsu, et al. "Bootstrapped training of score-conditioned generator for offline design of biological sequences." Advances in Neural Information Processing Systems 36 (2024).

[3] Trabucco, Brandon, et al. "Design-bench: Benchmarks for data-driven offline model-based optimization." International Conference on Machine Learning. PMLR, 2022.

**Questions:**

- Is there any performance degradation from not using the graph discovery heuristic? It would be helpful to compare the performance, at least on LRBF.
- Do you have further intuition on how to set $N_{\text{clique}} $ or $ d_{\text{clique}} $?

---

> ### Author Response · Authors · 2024-11-19
>
> ### Critique
>
> 1. The proposed architecture introduces new hyper-parameters ($N_{clique}$ and $d_{clique}$), and the authors claim that $d_{clique}=3$ consistently achieves good performance. However, the conducted ablation reveals that the model is sensitive to these hyper-parameters, and some settings of them can achieve bad performance.
>
> > The reviewer is right in pointing this out. Let us first clarify our statement about $d_{clique}=3$. As we described in *5.2 Ablations* and *Appendix B* (*Appendix C* in the revised version), in our experiments, we fixed $dim(z) = d_{knot} + N_{clique} \cdot (d_{clique} - d_{knot})$ to match $dim(x)$ as closely as we can, and thus by setting $d_{clique}=3$ and $d_{knot}=1$ we have also prescribed the value of $N_{clique}$. We suggest setting $d_{clique}=6$ for tasks with $dim(x) > 100$ to prevent computing attentions greater than $50\times 50$ - we can highlight this point in the final version. We indeed get good (though not always optimal) results with this scheme. Note that on 7 of our tasks, Cliqueformer uses these values, and on Superconductor where $3$ was sub-optimal, it does not underperform COMs that much (Figure 5b, $N_{clique}=20$).
> We can use this result if the reviewer suggests so.
>
> > The ablation study that the reviewer refers to, first of all, serves the purpose to prove that our novel mechanism that decomposes the latent variable into cliques does play a big role in the performance. We imagine that the reviewer agrees that failure to demonstrate this dependence would prove the mechanism useless. Indeed, the deep learning literature is abundant in valuable techniques that do not bring improvement under every single setting. For example, Dropout [1] can help generalization, but at a certain (mostly task-dependent) threshold, it leads to underfitting (see [1] Figure 9). Similarly, the NeurIPS-published work that the reviewer cited work on BDI reports in its ablation that removing one of their losses $L_{l2h}$ and $L_{h2l}$ can sometimes improve the performance. Thus, we kindly ask the reviewer to re-consider this comment.
>
> 2. The authors claim that Cliqueformer achieves state-of-the-art performance while some of the recent baselines have not been compared against.
>
> > We agree with the reviewer that such a claim requires an enlarged list of baselines. In the updated version, we added a comparison against recent diffusion-based DDOM [2], as well as gradient ascent with Transformer backbone, to provide further evidence that the performance of our model does not solely come from the backbone. The results strengthen evidence of our claim. We apologize for our prior negligence. Due to issues with DesignBench benchmark described in Appendix B (now, Appendix C), we had to spend extra time searching for alternative data which left us with less time to implement baselines. We settled on COMs due to its strong performance even against more recent methods. We will be extremely grateful if the reviewer takes this into consideration while re-examining our paper.
>
> ### Questions
>
> 1. Do you have any intuition on how to set $N_{clique}$ and $d_{clique}$?
>
> > This is a very valid question. As of now, we can observe what we stated in the paper, which is that $dim(z) \approx dim(x)$ and setting $d_{clique}=3$, or $d_{clique}=6$ for larger designs, tends to work well. We belive that the reason for that is that having more cliques - and thus smaller cliques - allows the decoder to learn more expressive attention patters, similarily to what was found in Diffusion Transformer [3] (Figure 6 bottom row). However, the cliques shouldn't be too small (like size 1) to enable more expressivity in the predictive model too. Our choice seems to find a sweet spot in this trade-off. A large-scale study of these hyper-parameters, both empirically and theoretically, is an exciting avenue of future work.
>
> ### References
>
> [1] N. Srivastave et al. 2014; Dropout: A Simple Way to Prevent Neural Networks from Overfitting
>
> [2] S. Krishnamoorthy et al. 2023; Diffusion Models for Black-Box Optimization
>
> [3] W. Peebles and S. Xie 2023; Scalable Diffusion Models with Transformers

---

> ### Comment · Reviewer_98z9 · 2024-11-25
>
> Thank you for your effort, and I apologize for the delayed response.
>
> **Critique**
>
> 1-a. Your responses have addressed my concerns regarding the hyperparameters. However, in the current version, I found it somewhat challenging to compare the performances presented in Figure 5 and Table 1. It would be helpful to include a brief comparison with the results from COMs either in the figures, captions, or main text. Additionally, the text in the figures is quite small, which makes it difficult to read.
>
> 1-b. I agree that the ablation studies don't need to demonstrate improvement in every setting. Nonetheless, I believe the comparison between $N_{\text{clique}}=1$ and $N_{\text{clique}}>1$ is important to understand the proposed method (though I don't expect Cliqueformer to outperform in every setup). Do you have any intuition for the results? For instance, I expected that using more cliques would be beneficial for high-dimensional tasks, but the results show that increasing $N_{\text{clique}}$ tends to give better results in TFBind-8 and lower scores in Superconductor.
>
> 2. Thank you for including a new baseline despite the limited time. It might be better to moderate expressions like "outperforms all baselines" in favor of something less strong than "state-of-the-art performance." Again, I don't mean to suggest that the performance of Cliqueformer is insufficient.
>
> **Questions**
>
> 1. Thank you for the clarification.
>
> I will consider raising my score if my remaining concerns in 1-a are addressed.

---

> ### Author Response · Authors · 2024-11-26
> **Further discussion**
>
> Thank you for your continuing work in helping us improve the paper. We are excited to respond to your suggestions and questions. We just submitted a revised version accordingly.
>
> 1a.
>
> > To improve the readability of Figure 5, we made new figures, in which we increased the font size of axis labels. We hope the figures are more legible now.
> The Reviewer asked about the relation of our results from Table 1 and the scores from the COMs paper. The key difference is that, in COMs, the reported results are in their raw form, without normalization (in the ICML version - in the Arxiv version, they are normalized like in DesignBench). This might make interpretation of the results difficult (e.g., what does it mean to score 333.4 in DKittyMorphology?), and that's why newer papers, like DesignBench (Trabucco et al. 2022) and DDOM (Krishnamoorthy et al. 2023), normalize their scores.
> To provide more clarity in reading the results in Table 1, in the caption under the table, we explain the reported scores as:
>
> > *The values were normalized with the min-max scheme, where the minimum and the maximum are taken from the dataset, so that the scores of the designs in the dataset are in range $[0, 1]$. We note that, unlike Trabucco et al. 2022, we take the maximum from the data available for the MBO model training (the union of the train and test data), and not from the oracle training data, to make the results more interpretable (see Appendix C).*
>
> 1b.
>
> > We agree with the reviewer that these observations seem tricky. For now, we can offer the following intuition: when we increase the number of cliques, we end up having a more expressive attention layer in the decoder, which can result in improved performance. On the other hand, note that the increased number of cliques implies more aggressive decomposition assumptions on the target function $f(z)$, potentially limiting the expressivity of our predictive model. We believe that, for different functions, there is a threshold into how many sub-functions it can be decomposed. Thus, we should expect the performance to improve until that threshold, and then start degrading once it's reached. Thus, we think that the shape of the curve in Figure 5 for TFBind8 is as expected. Probably, we did not get the same shape in Figures (b) and (c) because of the large exponential leaps in the number of cliques tested. Note that we made those leaps because we needed $N_{clique}$ to divide $dim(z) - d_{knot}$ (see Appendix C, Line 860).
>
> 2.
>
> > We appreciate the reviewer's suggestion about the most proper choice of words. We have replaced the "state-of-the-art performance" with "outperforms the baselines" in both abstract and the experiment section. Does that seem like a reasonable change?

---

> > ### Comment · Reviewer_98z9 · 2024-11-26
> >
> > Thank you for the additional work. I've increased my score to 6, which indicates a positive inclination toward acceptance.

---

### Author Response · Authors · 2024-11-19

We thank all the reviewers for the time they took to read our paper and provide feedback that we are using to turn our paper into a better shape. We have carefully reviewed your comments and addressed most prompting of your requests, specifically regarding additional experiments. We also did our best to provide clarifications to the posed questions, and we hope for a fruitful discussion that can further improve the paper's clarity. Our responses follow format
### Critique
> where we list and answer all points of critique from the reviewer

### Questions
> where we list questions raised by the reviewer and do our best to answer them.

We believe that our effort will address the confusion regarding the position of our paper of reviewers **eafL** and **JStj**, and if so, result in increased scores of our paper.

---

### Author Response · Authors · 2024-11-22
**Next steps**

Dear Reviewers,

We believe we have addressed your concerns. As the next step, we would appreciate your feedback during this discussion period. Should you have any other concerns, we are constantly ready to address them next. Let us know if there is anything else we should do.

Thanks you,
Authors

---

### Meta-Review · Area_Chair_NBnQ · 2024-12-20

**Metareview:**

This paper introduces Cliqueformer, a new architecture for model-based optimization (MBO) problems such as chemical and genetic design tasks. The proposed model aims to improve out-of-distribution generalization by learning and predicting a functional graphical model of the underlying black-box function. The field of model-based optimization is relatively new, so there is potential for this method. However, this paper would greatly benefit from another peer review before it is accepted for publication. The authors performed limited experiments, few ablation studies focused on hyperparameter sensitivity and did not describe computational complexity. Most of the paper (the first six pages) focuses on intuitions and desiderata of MBO methods. I suggest that the authors use the space in their paper more strategically, dedicating more space to results, hyperparameter studies, and *experimental evidence* as to why their approach is significant and of interest to the community.

**Additional Comments On Reviewer Discussion:**

During the rebuttal period, the authors and reviewers discussed the paper extensively. Many of the reviewers expressed concerns about relevant related work (both references and experimental baselines), limited experiments, and limited evidence through ablations and hyperparameter sweeps that justified the intuitions laid out earlier in the paper. Though the authors did include additional comparisons and discussions, most of their responses to reviewers were defensive, often citing that asks from reviewers were not "unique flaw[s] of our paper" or claiming that seminal papers like AlexNet and Transformer did not perform ablation studies.

The authors missed an excellent opportunity to improve their work by incorporating reviewer feedback. Even if this paper had been accepted as is (or accepted to a future conference as is), it could have been significantly more substantial and impactful if the authors included additional benchmarks or experimental evidence that justified their intuitions.

I agree with the authors that MBO is a relatively new field that can make the paper challenging to review. Nevertheless, I generally agree with the reviewers' concerns and hope the authors sincerely consider them as they revise their work.

---

### Decision · Program_Chairs · 2025-01-22

Reject